# Spatially Extensive Long-term Quality-assured Land-atmosphere Interactions Dataset over the Tibetan Plateau

Yaoming Ma[1,6,13,14,15,16], Zhipeng Xie[1,6*], Yingying Chen[1, 2*], Shaomin Liu[3], Tao Che[4], Ziwei Xu[3], Lunyu Shang[5,7], Xiaobo He[6], Xianhong Meng[5,7], Weiqiang Ma[1,6,14], Baiqing Xu[1,9,13], Huabiao Zhao[1,10], Junbo Wang[1,11], Guangjian Wu[1,12], Xin Li[1,2,13*]

[1] State Key Laboratory of Tibetan Plateau Earth System, Environment and Resources (TPESER), Institute of Tibetan Plateau Research, Chinese Academy of Sciences, Beijing 100101, China

[2] National Tibetan Plateau Data Center, Institute of Tibetan Plateau Research, Chinese Academy of Sciences, Beijing 100101, China

[3] State Key Laboratory of Earth Surface Processes and Resource Ecology, Faculty of Geographical Science, Beijing Normal University, Beijing 100875, China

[4] Heihe Remote Sensing Experimental Research Station, Key Laboratory of Remote Sensing of Gansu Province, Northwest Institute of Eco-Environment and Resources, Chinese Academy of Sciences, Lanzhou 730000, China

[5] Key Laboratory of Land Surface Process and Climate Change in Cold and Arid Regions, Northwest Institute of Eco-Environment and Resources, Chinese Academy of Sciences, Lanzhou 730000, China

[6] National Observation and Research Station for Qomolangma Special Atmospheric Processes and Environmental Changes, Dingri 858200, China

[7] Zoige Plateau Wetland Ecosystem Research Station, Northwest Institute of Eco-Environment and Resources, Chinese Academy of Sciences, Lanzhou 730000, China

[8] Tanggula Mountain Cryosphere and Environment Observation and Research Station of Tibet Autonomous Region, Northwest Institute of Eco-Environment and Resources, Chinese Academy of Sciences, Lanzhou 730000, China

[9] Muztagh Ata Station for Westerly Environment Observation and Research, Institute of Tibetan Plateau Research, Chinese Academy of Sciences, Beijing 100101, China

[10] Ngari Station for Desert Environment Observation and Research, Institute of Tibetan Plateau Research, Chinese Academy of Sciences, Beijing 100101, China

[11] Tibet Nam Co High-cold-lake and Environment National Monitoring Observation and Research Station, Damxung 851500, China

[12] South-East Tibetan Plateau Station for Integrated Observation and Research of Alpine Environment, Institute of Tibetan Plateau Research, Chinese Academy of Sciences, Beijing 100101, China

[13] College of Earth and Planetary Sciences, University of Chinese Academy of Sciences, Beijing 100049, China

[14] College of Atmospheric Science, Lanzhou University, Lanzhou 730000, China

[15] Kathmandu Center of Research and Education, Chinese Academy of Sciences, Beijing 100101, China

[16] China-Pakistan Joint Research Center on Earth Sciences, Chinese Academy of Sciences, Islamabad 45320, Pakistan

*Correspondence to*: Zhipeng Xie (zp_xie@itpcas.ac.cn), Yingying Chen (chenyy@itpcas.ac.cn), Xin Li (xinli@itpcas.ac.cn)

**Abstract.** The climate on the Tibetan Plateau (TP) has experienced substantial changes in recent decades as a result of its susceptibility to global climate change. The changes observed across the TP are closely associated with regional land-atmosphere interactions. Current models and satellites struggle to accurately depict the interactions, critical field observations on land-atmosphere interactions here therefore provide necessitate independent validation data and fine-scale process insights for constraining reanalysis products, remote sensing retrievals, and land surface model parameterizations. Scientific data

sharing is crucial for the TP since in-situ observations are rarely available in this harsh condition. However, field observations are currently dispersed among individuals or groups and have not yet been integrated for comprehensive analysis. This has prevented a better understanding of the interactions, the unprecedented changes they generate, and the substantial ecological and environmental consequences they bring about. In this study, we collaborated with different agencies and organizations to present a comprehensive dataset for hourly measurements of surface energy balance components, soil hydrothermal properties, and near-surface micrometeorological conditions spanning up to 17 years (2005-2021). This dataset, derived from 12 field stations covering a variety of typical TP landscapes, provides the most extensive in-situ observation data available for studying land-atmosphere interactions on the TP to date in terms of both spatial coverage and duration. Three categories of observations are provided in this dataset: meteorological gradient data (Met), soil hydrothermal data (Soil), and turbulent flux (Flux). To assure data quality, a set of rigorous data processing and quality control procedures are implemented for all observation elements (e.g., wind speed and direction at different height) in this dataset. The operational workflow and procedures are individually tailored to the varied types of elements at each station, including automated error screening, manual inspection, diagnostic checking, adjustments, and quality flagging. The hourly raw data series, the quality-assured data, and supplementary information including data integrity and the percentage of correct data on a monthly scale are provided via the National Tibetan Plateau Data Center (https://doi.org/10.11888/Atmos.tpdc.300977, Ma et al., 2023a). With the greatest number of stations covered, the fullest collection of meteorological elements, and the longest duration of observation and recording to date, this dataset is the most extensive hourly land-atmosphere interactions observation dataset for the TP. It will serve as the benchmark for evaluating and refining land surface models, reanalysis products, and remote sensing retrievals, characterizing fine-scale land-atmosphere interaction processes of the TP, as well as underlying influence mechanisms.

## 1 Introduction

The Tibetan Plateau (TP) is the most spatially extensive highland with the highest altitude in the world. Strong interactions occur among nearly all major Earth system components, including the lithosphere, atmosphere, cryosphere, biosphere and anthroposphere, making it an ideal natural laboratory for studying the Earth system (Yao et al., 2019; Chen et al., 2021). However, the climate over the TP is highly sensitive to global climate change owing to its distinct terrain and geographical location, with warming amplified compared to other regions (Liu and Chen, 2000). It is therefore characterized as an amplifier for global climate change (Pan et al., 1996). Many of the TP's environmental system components have experienced evident changes over the past few decades (Kang et al., 2010), for example, substantial retreat of glaciers (Yao et al., 2012), and dramatically decrease of snow depth and snow cover under the warming climate (Qin et al., 2006; You et al., 2020). These consequences together provide crucial feedback that exacerbates warming and hydroclimatic changes across the TP. Emerging evidence indicates that critical climate tipping points are likely being rapidly approached over the TP due to rising global temperatures, and passing through these thresholds could potentially have significant social consequences (Armstrong et al., 2022).

Under these circumstances, policymakers and stakeholders rely heavily on the scientific research community to develop reliable models that can assist in designing and implementing effective forward-looking mitigation and adaptation strategies to help combat the effects of climate change (Thornton et al., 2021). However, developing these models is challenging and necessitates a thorough physical understanding on how climate system operates and interacts, which relies on acquiring informative, comprehensive, and representative environmental data. Meanwhile, regional land-atmosphere interactions are closely linked to environmental system changes (Zhou et al., 2019), even if anthropogenic climate forcing is recognized as the primary driver of these changes (Duan et al., 2006). In particular, quantifying energy and water exchanges across the heterogeneous landscape of the TP remains a challenge for the scientific community (Ma et al., 2023c). Thus, it is imperative to gain a thorough understanding of the multiscale land-atmosphere interactions taking place over the TP. Establishing appropriate countermeasures in response to global climate change and to offering insights into the associated ecological and environmental vulnerability are both made possible by this.

The TP presents immense challenges for the research on land-atmosphere interactions and feedback, arising from the inherent complexities of such interactions. However, challenges in obtaining crucial observations impede an enhanced comprehension of these interactions work, the unprecedented changes they generate, and the massive ecological and environmental consequences they pose. In addition, site selection is made more difficult by the complex mountainous topography, which also raises issues of spatial representativeness and calls for dense station networks. Furthermore, a great deal of important processes is inherently linked by intricate feedback loops. The considerable diversity and spatiotemporal heterogeneity of these interactions complicate the modelling of water and heat fluxes transfer across this mountainous region. Data scarcity has constrained research on coupled land-atmosphere dynamics in this sensitive region. While remote sensing helps to fill in certain observation gaps and provides some insights into surface properties, space-borne sensors struggle with complex land surfaces and frequent cloud cover. Therefore, in-situ measurements remain essential to elucidate fine-scale exchange processes and to validate retrievals. Although altitude, terrain, isolation, and climate pose combined challenges for establishing, maintaining, and managing the in-situ network indispensable for monitoring the land-atmosphere interactions over the TP, expanding the observation network is imperative to fill in knowledge gaps.

In recent decades, significant progress has been made in establishing comprehensive field stations to comprehend the mechanisms of the regional water and energy cycles and provide long-term observations for land-atmosphere interactions across the TP. This has been made by major field campaigns, large-scale scientific experiments, and infrastructure development under research programs. After decades of effort, the Tibetan Observation and Research Platform (TORP, Ma et al., 2008), the Third Pole Environment (TPE) Observation and Research Platform (TPEORP, Ma et al., 2018), and the TPE Integrated Three-dimensional Observation and Research Platform (TPEITORP, Ma et al., 2023c) have been established with optimized scientific design and layout. These efforts have expanded the availability of in-situ meteorological, hydrological, and cryospheric measurements. For example, 29 comprehensive stations in the TPEITORP are dispersed throughout the major ecosystems and climatic gradients of the TP (Ma et al., 2023c). This makes it possible to characterize the heterogeneous water, energy, and carbon exchanges associated with the complex terrain. A large amount of valuable long-term observations has

been obtained and these observations have been widely used in a range of disciplines and spatial scales, greatly advanced our understanding of the land-atmosphere interactions, and supported the development of land surface models and facilitated the refinement of satellite retrieval algorithms (Yuan et al., 2021; Ma et al., 2023c).

A rising number of field observation datasets are progressively being made accessible and freely available to the public (e.g., Che et al., 2019; Ma et al., 2020; Liu, et al., 2023; Meng et al., 2023). However, to the best of our knowledge, thorough data quality control procedures have not been implemented to the current publicly available datasets for land-atmosphere interactions over the TP, although these datasets have contributed significantly to the study of climate and environmental change in the mountainous region. The extreme conditions over the TP pose numerous threats to sensor functioning and can introduce measurement errors. These factors can degrade calibration or alter instrument response characteristics, resulting in biased readings. With sparse monitoring networks, even small data problems can propagate into significant uncertainties when datasets are used for analysis or assimilated into models. Unfortunately, uncertainties in observations at many different stages of their life cycle pose a barrier to guaranteeing the high quality of the field observations (Fiebrich et al., 2010). Although proactive maintenance and sensor recalibration can greatly improve data quality, errors may still be inevitable. Data quality is one of the main concerns in scientific studies, particularly in the TP. These risks require mitigation through rigorous quality control procedures to ensure the reliability, consistency, and integrity of the observation datasets. An effective quality assurance system has been recognized as essential for producing reliable, high-quality data (Peppler et al., 2008). While there are alternative software packages available for data quality assurance (e.g., MetPyQC, and MADIS), our experience is that no single algorithm or combination of them can identify all the specific data quality problems that exist at each field station, as data problem varies considerably from one field station to another. Therefore, it is imperative to develop specific automatic data post-processing algorithms to handle the unique problems arising from each station.

With regard to observation for land-atmosphere interactions over the TP, current available in-situ data are restricted to short intense campaigns or cover only a few sites or particular processes, and they ignore data uncertainties brought on by possible sensor noise or measurements fluctuations caused by changes in weather conditions. This work closes these important gaps in the literature. The dataset presented here includes simultaneous measurements of surface energy balance components, soil hydrothermal properties, and near-surface micrometeorological conditions compiled from 12 field stations affiliated with different organizations, enabling a holistic analysis of land-atmosphere feedback. This represents the most extensive in-situ observation dataset of land-atmosphere interactions available for the TP to date in terms of spatial coverage and duration. Sets of automated post-processing algorithms and manual assessment are specifically designed for the multi-process observation network of land-atmosphere interactions. This dataset undergoes a series of robust quality checks, including automated error screening, manual inspection, diagnostic checking, adjustments, and quality flagging to guarantee that any suspicious observations are identified. By compiling this unique collection of multi-year, multi-variable in-situ observations across the TP and making it freely available to the scientific community, this work aims to advance process-level understanding of the land-atmosphere interactions over this climatically sensitive region and reduce uncertainties in satellite remote sensing, weather and climate predictions.

## 2 Observation network and data processing

### 2.1 Integrated land-atmosphere interactions observation network

This dataset compiles integrated land-atmosphere interaction observations from twelve stations (Fig. 1), ranging from 3033 to 5150m in altitude (Table. 1). All the sites are characterized by flat and open terrain with a relatively homogeneous underlying surface surrounded, making them an ideal experimental field for observing and investigating the land-atmosphere interactions in varied topography of the TP region.

Four sites (Arou, Dashalong, Jingyangling and Yakou) are in the upper reaches of the Heihe River (UHR) basin, an endorheic basin with arid and semiarid climate located in the northeast of the TP (Liu et al., 2018). The vegetation and soil type at these sites are dominated by alpine meadow and silt loam. The Arou station was constructed in a valley to the south of Babao River, a tributary of the UHR, surrounded by rather flat and open terrain. The Yakou station is located on the highland on the east side of Dadongshu Mountain in the UHR. It is an ideal snow observation station as the experimental area is frequently blanketed with snow in autumn, winter, and spring. The annual precipitation at Arou station and Dashalong station was approximately 570 and 420 mm, respectively, while it reached 600-800 mm at the Yakou station (Liu et al., 2018).

Two sites (Ngoring Lake and Maqu) are located in the source region of the Yellow River Basin, an "extremely sensitive region" to climate change and has a semi-arid climate (Lu et al., 2018; Xu et al., 2020). The Ngoring Lake site was established on a relatively flat grassland, about 2 km west of Lake Ngoring (Meng et al., 2023). A height of about 0.05-0.1 m alpine meadow covers its underlying surface. The Maqu site is situated in south of the Maqu County, characterized by alpine meadow with a height of 0.2 m in the summer and about 0.1 m in the winter (An et al., 2020). The Maqu site receives roughly 600 mm of precipitation on average, most of which falls during the summer because of the summer monsoon, and snowfall occurs very infrequently during the winter (Wang et al., 2017).

The TGL site is located in the Dongkemadi River Basin in the source region of the Yangtze River on the TP, equipped with the world's first Eddy covariance system on the uniform alpine swamp meadow above 5,000 m (Guo et al., 2022).

The other five sites are all affiliated with the Institute of Tibetan Plateau Research, Chinese Academy of Sciences (ITPCAS). The National Observation and Research Station for Qomolangma Special Atmospheric Processes and Environmental Changes (QOMS) is situated at the bottom of the lower Rongbuk Valley, 30 km to the north of Mt. Qomolangma, is an ideal location for monitoring atmospheric conditions in the Northern Hemisphere. Situated at the southeastern shoreline area of Nam Co (Co means lake), the Tibet Nam Co High-cold-lake and Environmental National Observation and Research Station (NAMORS), is an ideal place to measure the land-atmosphere interactions in the lake-land-mountain mesoscale system. Situated between Mt. Muztagh and Lake Karakuri, the Muztagh Ata Westerly Observation and Research Station, CAS (MAWORS) is a typical westerly climate zone with year-round westerly influences. The Ngari Station for Desert Environment Observation and Research, CAS (NASED), is ideally situated to study the westerly-monsoon interactions on the desert landscape, as it is located at the convergence zone of the Indian monsoon and westerly. In a mountain valley near the forested southeast TP is the South-East Tibetan Plateau Station for integrated observation and research of alpine

environment, CAS (SETORS). It is an important site for tracking the water and heat transport along the alpine valleys by the South Asian monsoon. For further details on these five sites, please refer to Ma et al., (2020).

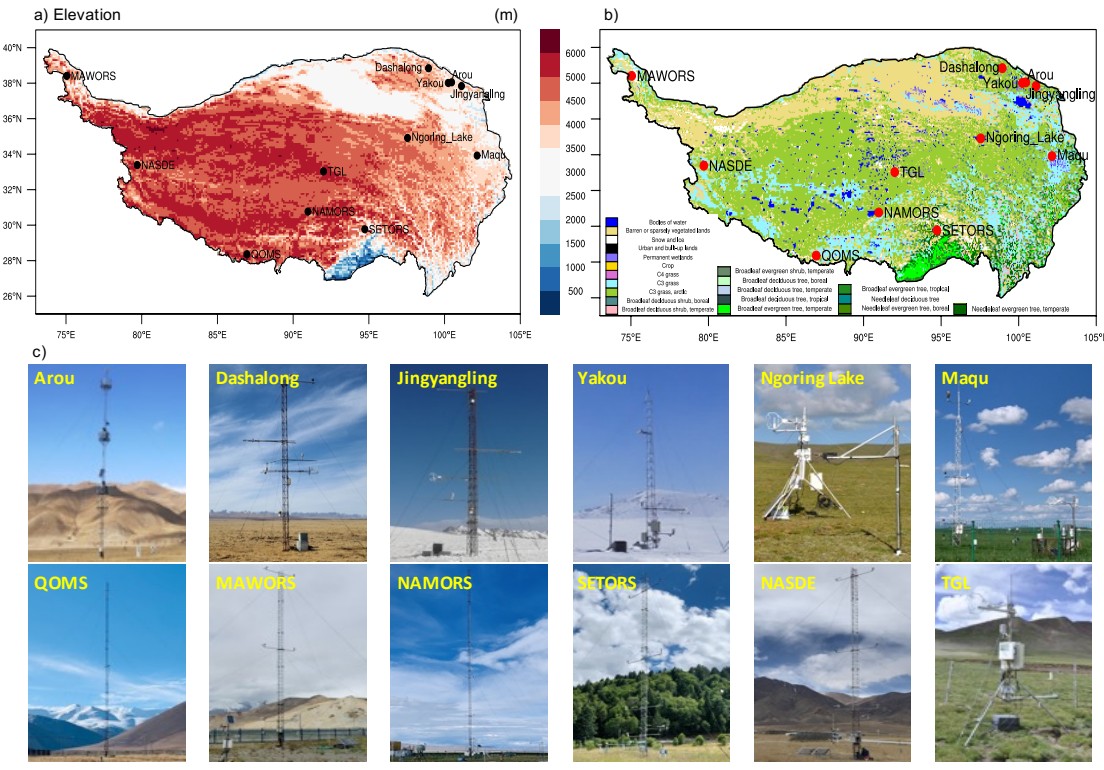

Figure 1. (a) The elevation, (b) land use and cover classification, and (c) photos of the integrated land-atmosphere interactions observation network.

## 2.2 Observation infrastructure

The integrated observation network for land-atmosphere interactions includes six superstations (after upgrading and reconstruction for MAWORS and NASED in 2020) with multi-layer meteorological data (up to 6 layers at the Arou station) and six ordinary stations that measure single layer near-surface meteorological elements (Table. 1). These stations cover the main landscapes of the TP, including alpine meadows, deserts, steppes, and grasslands. Vertical profiles of meteorological conditions (e.g., wind speed and direction, air temperature, and relative humidity) within the lower boundary layer are measured for the superstations (Arou, MAWORS, NASED, NAMORS, QOMS, and SETORS), but with different planetary boundary layer (PBL) tower observing system layout (e.g., sensor height, and instrument model).To ensure consistency, it is important to emphasize that all instruments deployed at NAMORS, QOMS and SETORS before 2020 and all five stations affiliated with ITPCAS (MAWORS, NASED, NAMORS, QOMS, and SETORS) since 2020 follow a similar deployment strategy. For instance, five layers of wind speed and direction anemometers, air temperature and humidity probes, as well as one layer of air pressure barometer, rain gauge, and four-component radiation sensor were deployed at each station. These

instruments are of the same model at the five sites since the reconstruction of the observing system in 2020 and were installed

at comparable or identical heights whenever feasible (for more specific information on site deployment, see Table. 1). While these near-surface meteorological components were observed at one level for regular stations (TGL, Ngoring Lake, Maqu, Yakou, and Jingyangling) as well as the MAWORS and NASED stations before reconstruction.

Vertical soil hydrothermal profile of each station was measured with multilayer soil temperature-moisture probes. Except for a few sites (MAWORS, NASED, NAMORS, QOMS, and SETORS) where the depth of thermal sensors and probes for

measuring soil hydrothermal properties was constant, there were notable variations in the number of layers and depth at the other sites. Although soil heat flux was also measured at the field stations constructed by ITPCAS using soil heat flux plates (HFP01SC, Hukseflux, Netherlands), observations are not included in this dataset because its reliability cannot be fully guaranteed based on our assessment (Ma et al., 2020). Detailed information regarding the sensors used and the heights the sensors laid is listed in Table 1.

A high-frequency eddy covariance (EC) system was deployed at each site to capture the sensible heat, latent heat, and carbon dioxide fluxes. The EC system consists of a sonic anemometer (CSAT3/CSAT3B, Campbell Scientific, USA) and a fast-response infrared gas-analyzer (LI-7500/7500A/7500DS/7500RS, LI-COR, USA). While for Maqu, the EC150 open-path gas analyzer (Campbell Scientific, USA) was used. The turbulent exchange of heat, water vapor, carbon dioxide, and momentum were measured at 10 Hz. The high frequency data series underwent quality verification and processing at 30-

minute average intervals. Specific processing procedures will be described in section 3.1. A specific correction to the carbon dioxide flux observations is highly needed but has not yet been done, since the self-heating issue of the instruments can have a significant impact on the measurement accuracy and stability of the open-path infrared gas-analyzer (Burba et al., 2008). Thus, this dataset does not include the carbon dioxide flux. The carbon dioxide flux observations will also be publicly shared after the data correction and validation are finished.

Calibration of instruments is critical for ensuring accurate measurements. It is important to note, however, calibrating in a particularly harsh environment such as the TP is challenging. As a result, for meteorological and soil observations, both of which are relatively stable, calibrated reference instruments were used on a regular basis to perform field calibration across multiple stations, or the calibration was performed in a laboratory setting when instruments were returned for repair. In the case of turbulent observations, the measurement accuracy of the gas analyzer (i.e., LI-7500 and LI-7500DS) depends upon the

cleanliness of the instrument lenses, it needs to be calibrated at regular intervals (once every six months at the five sites affiliated with the ITPCAS) due to signal attenuation for CO2/H2O. The calibration consists of two major components: 1) determining the values of the calibration coefficients, and 2) adjusting zero and span to align the gas analyzer's actual response with the previously determined factory response. In addition, we conduct monthly inspections of the operational status of all observational equipment (Ma et al., 2023b), as well as semi-annual on-site instrument maintenance for all stations, which

includes instrument cleaning, checking the level of commissioned instruments, and checking instrument cables and connectors.To the maximum extent feasible, qualified personnel will take over and rectify any instrument malfunctions found during routine inspection (on-site or remote) to ensure the accuracy and integrity of the observations. Data logger (e.g., CR6,

Campbell Scientific, USA) recordings are first temporarily stored on the memory card before being routinely transmitted to our Data Processing Center by wireless transmission or on-site collection for processing, analysis, and archiving.

**a). Post-processing workflow**

| Raw observations | 1. Data processing | 2. Quality control | 3. Gap filling | 4. Data archiving |
|---|---|---|---|---|
| **Meteorological data**<br>■ Wind speed (WS)<br>■ Wind direction(WD)<br>■ Air temperature(Ta)<br>■ Relative humidity(RH)<br>■ Air pressure(Pressure)<br>■ Downward shortwave radiation(Rsd)<br>■ Upward shortwave radiation(Rsu)<br>■ Downward longwave radiation(Rld)<br>■ Upward longwave radiation(Rlu)<br><br>**Soil data**<br>■ Soil temperature(ST)<br>■ Soil moisture(SM)<br><br>**10Hz turbulent data**<br>■ Ux<br>■ Uy<br>■ Uz<br>■ T_sonic<br>■ $CO_2$ density<br>■ $H_2O$ density | ■ **Spike detection***<br>■ **Sonic temperature correction***<br>■ **WPL correction***<br>■ **Turbulent flux calculation***:<br> Sensible heat flux (H)<br> Latent heat flux (LE)<br>■ **Diagnostics for missing times:**<br> WS, WD, Ta, RH, Pressure, Rsd, Rsu, Rld, Rlu, H, LE<br>■ **Missing value assignment:**<br> WS, WD, Ta, RH, Pressure, Rsd, Rsu, Rld, Rlu, H, LE<br>■ **Format conversion:**<br> • Meteorological files<br> • Soil files<br> • Flux files<br><br>* Raw turbulent data only | ■ **Range checks:**<br> WS, WD, Ta, RH, Pressure, Rsd, Rsu, Rld, Rlu, ST, SM<br>■ **Temporal consistency checks:**<br> • Persistence tests:<br> WS, Ta, RH, Pressure<br> • Step tests:<br> WS, WD, Ta, RH, Pressure, Rsd, Rsu, Rld, Rlu<br>■ **Internal consistency checks:**<br> WS&WD, ST(different depth)<br>■ **Expert quality assessment:**<br> WS, WD, Ta, RH, Pressure, Rsd, Rsu, Rld, Rlu, ST, SM, H, LE | ■ **Short gap filling*:**<br> WS, WD, Ta, RH, Pressure, Rsd, Rsu, Rld, Rlu, ST, SM<br><br>* Gap filling is performed only if there are no more than 3 consecutive missing data | ■ **Standardized data header description**<br>■ **Standardized data file**<br> • Meteorological files(Met)<br> • Soil files(Soil)<br> • Flux files(Flux)<br>■ **Data aggregation** |

**b). Quality control**

| Range checks | Temporal checks | Internal consistency checks | Manual quality assessment |
|---|---|---|---|
| $QC=\begin{cases}0,\ Min \le Obs \le Max* \\ 2,\ Obs < Min\ or\ Obs > Max\end{cases}$<br><br>* Range of limits for each of the observed variables are listed in Table 2 | **Persistence tests**<br>QC= 1, $|Obs_i-Obs_{i-1}|$ <threshold $\Delta x$*<br>QC= 2, values remain unchanged for more than 24 consecutive hours<br><br>**Step tests**<br>QC= 2, $|Obs_i - run\_avg(7)| > 3 \times STD(7)^\$$<br><br>* Threshold $\Delta x$ is defined at hourly scale, and it is listed in Table 2;<br>$ run_avg(7) is the moving average with a window size of 7 hours, STD(7) is the corresponding standard deviation. | ■ For WS and WD only:<br> QC= 1, WS > 0 and WD = 0;<br>  WS = 0 and WD > 0.<br>■ For ST only:<br> Check was performed based on adjacent layers of ST series<br> • QC=2, $|Obs_i- 0.5 \times (Obs_{i-1}-Obs_{i+1})| \ge 3.0$ for shallow layer ST;<br> • QC=2, $|Obs_i- 0.5 \times (Obs_{i-1}-Obs_{i+1})| \ge 4.0$ for deep layer ST. | ■ Assessing variation in minimum, average, maximum, and standard deviation of each variable<br> • seasonal diurnal cycles;<br> • long-term variation;<br> • values at adjacent heights/depths |

**Figure 2.** a) Flowchart of the data post-processing workflow, and b) the schematic diagram of the processing chain of the quality checks. Main steps are printed in red.

## 2.3 Data post-processing workflow

To compile a high-quality set of in-situ land-atmosphere interaction observations, this study used a series of automated post-processing algorithms and manual assessment specifically designed for this multi-process observation network of land-atmosphere interactions. The data processing workflow can be divided into data processing, quality control, gap filling, and data archiving (Fig. 2).

### 2.3.1 Data processing

The first step of data processing was mostly tailored to the raw 10 Hz turbulent data, while several operations (e.g., diagnostics for missing times, and format conversion) were applied to all variables. All raw turbulent data were subjected to spike detection, spike and trend removal, sonic temperature correction, coordinate rotation, frequency response correction, and WPL (Webb-Pearman-Leuning) correction, in accordance with the procedures implemented by Ma et al., (2020), Meng et al., (2023), and Liu et al., (2023). The quality of each specific turbulent flux value was evaluated according to the steady state test and developed turbulence conditions test. Flagging policy proposed by Mauder and Foken (2006) was used to combine the two flags from the two tests into a final quality flag: 0 presents the highest quality fluxes, 1 represents fluxes suitable for general analysis, and 2 represents fluxes that should be discarded. In addition to some targeted processing for the turbulent data, the continuity of the timestamps for all types of data was diagnosed. Although either NAN or -9,999 was used to denote a missing value prior to the post-processing of the data, the final data used 9,999.9 to indicate missing data. Additionally, different instruments collected and recorded the data on various data loggers, and different organizations employed various data formats to store structured data; for these reasons, merging the data into a common format is a crucial step in data processing. The plain ASCII comma-separated values (CSV) file format was used in this study.

### 2.3.2 Data quality control

Quality control is particularly demanding for micrometeorological observations in complex environments. Owing to the unique characteristics of the climate and the way the observation network for land-atmosphere interaction was configured, certain practical issues detected in our previously released dataset (Ma et al., 2020) were not adequately addressed by some of the traditional quality control techniques currently in use. To provide the best level of accuracy feasible, an automatic processing scheme was specifically designed for each type of variable, following the guidelines described by Zahumensky (2004). The guidelines provide comprehensive documentation on basic quality control procedures used to ensure the accuracy of meteorological observations following World Meteorological Organization (WMO) standards. Despite a wide array of methods has been proposed to obtain plausible micrometeorological data series, those methods share similar processing flow but varying degrees of modifications were made to deal with site-specific concerns and unique problems that emerged from the field observations. This is due to the fact that generic methods frequently failed to resolve these issues. This scheme is specifically adapted aimed at verifying the reliability of observations and detecting errors and suspicious values. The automatic data processing chain was built up as a series of sequential checks recommended by Zahumensky (2004), with emphasis on continuity and inter-consistency of meteorological fields to detect suspect observations.

The program first reads the data file for each station to be processed, checks were performed sequentially from left to right, and only when all the prescribed check procedures for each variable completed before moving on to the next one. The quality control procedures are arranged in a deliberate sequence, and ignore values flagged as errors by preceding checks in

the sequence because the checks each have specific data requirements (e.g., running average and corresponding standard deviation should be calculated based on correct data).

270     ***Range checks***: the purpose of this check is to confirm that the values fall into a plausible range. Outliers are frequently encountered in field observations, particularly in the harsh natural environment on the TP. The wide variety of outliers is typically caused by unanticipated abrupt changes in the surrounding conditions, instrument fluctuations, or data-logger malfunction. The data points that failed the check were flagged as errors (with a QC score of 2). The range limits used in this study were constant, ignoring the seasonal variations, in contrast to the climatological limit check, where the range of limits

275     depends on the season and regional climatic conditions (typically at least 20 years of archived data are required to define climate range thresholds). The constant limits were predetermined based on the sensor hardware specifications, theoretical limits, and following the limits in climatological conditions recommended by the Zahumenský (2004). Minor adjustments were made at each site based on its unique climate to guarantee accurate and reliable detection results. Table 2 provides the range of limits for each of the observed variables.

280     ***Temporal checks***: this check assesses the validity of changes in the time series of data at a station. Two tests were used to evaluate the temporal consistency: the persistence test was implemented to verify the "dead band" caused by blocked sensors or instrument malfunction by checking a minimum required variability between sequential values; and the step test was used to detect unrealistic jumps in values with a verified maximum allowed variability because of the changes in weather conditions. It was discovered during testing that there would be a significant uncertainty if soil temperature, soil moisture, and surface

285     radiations were subjected to temporal consistency test since the former two elements frequently exhibit little to no fluctuation but surface radiations are expected to exhibit substantial variability. As a result, we did not perform similar check for surface radiations, soil temperature, and soil moisture. This study used a more subjective set of thresholds (minimal and maximum values) compared with range tests (Table 2). For the persistence test, when target variable values remain unchanged for more than 24 consecutive hours, data is deemed erroneous, and corresponding QC flags are assigned to 2. The step test was

290     performed based on the standard deviation threshold. Moving average with a window size of 7 hours was used to calculate the standard deviation of the sub-data list and the difference between each value and the mean value of this data list. Values that were more than three standard deviations from the mean were thought to be potentially erroneous and thus were flagged with 2.

    ***Internal consistency checks***: to identify observations that deviate from a realistic meteorological relationship between

295     two parameters (or the same variable at different heights). When an internal consistency check was conducted, data with concurrent wind direction (wind speed) greater than 0 and zero wind speed (wind direction) were flagged as suspicious (with QC code assigned to 1). Furthermore, if the wind speed and direction values remained unchanged for 12 consecutive hours, the likely cause was either a sensor that seized up due to extremely cold temperatures as reported by Wang et al., (2023) or a data logger crash that was occasionally observed at some sites on the TP. In this study, we performed internal consistency

300     checks on soil temperature to verify their coordinated variation, making use of the relationships between adjacent layers of soil temperature series. Two layers that were adjacent to the target depth were employed as reference series. The data points

were marked as errors when the data value differed from the mean value of two reference values by more than the site-specific defined thresholds (determined by a layer-by-layer analysis of the soil temperature time series at each site). For the first (deepest) layer, two layers below (above) the target layer are reference series. For other layers, the reference series is the one layer above and below the target layer.

*Manual quality assessment*: expert quality assurance is a vital means of ensuring the data quality since it offers the crucial pieces of information that a quality assurance meteorologist needs to determine which data require further analysis. When all of the data points are within acceptable bounds and there isn't an abnormal change, the algorithm might fail to accurately diagnose incorrect data. For instance, during the period of August 2-5, 2017, the maximum daily air temperature at the Dashalong station was approximately 30 degrees, which was significantly higher than the mean maximum temperatures for the same period historically and on neighbouring days (Fig. 3a and b). Unfortunately, it was not possible to identify and flag these abnormally high temperatures (Fig. 3b) as these values were within the acceptable range. Further assessment is necessary to detect these abnormal values. Therefore, the seasonal diurnal cycles, the long-term variation in minimum, average, maximum, and standard deviation of each variable at the daily scale were carefully examined. The reliability of the suspicious values was assessed by comparing them with other relevant variables (or the same variable but at different heights/depths) before performing manual QC code adjustments on these data points (see Fig. 3d and e respectively for comparing the abnormal temperature series at 1.3 m height from February 2017 through May 2018 with multi-year monthly mean and daily mean observed at 4.94 m, and Fig. 3f and g for the abnormal variations of downward/upward longwave radiation in 2015 and 2021 for the Arou station). Table 3 lists all the QC scores that were manually revised following expert quality assessment.

### 2.3.3 Gap filling

For all the data points marked as suspect, erroneous, and missing values, the linear temporal interpolation method was applied if the number of consecutive non-zero QC flags was less than 3. Only short gaps were interpolated because the performance of the reconstruction method is strictly dependent on the length of the data gap (Henn et al., 2013). We believe that the entire set of procedures proposed here serves primarily to detect errors and was not intended to reconstruct data, although sophisticated methods (e.g., probabilistic filling and artificial neural network) may offer increased performance for long-missing gap filling, a simple but effective linear interpolation method was used. All the gap-filled data points have their QC flag adjusted to 4. Series of random gaps (5,000 records for each variable) with different lengths were artificially created to quantify the overall performance of the gap filling method used and the robustness of the gap-filled data produced. The performance in filling gaps in wind speed, wind direction, air temperature, relatively humidity, downward/upward shortwave radiation, soil temperatures (at depths of 0.1 and 0.8 m), sensible heat flux, and latent heat flux for gap lengths of 1, 2, 3 hours were evaluated. These variables were selected for assessment because they exhibit varying degrees of variability in the observed values over relatively short intervals. For example, wind speed and wind direction vary significantly over 1-3 hours, whereas soil temperature changes less during that time. Table 4 shows the mean error, mean absolute error, root mean square

error, p value from the t test, and r square. Results suggest that the gap length is one of the key factors influences the performance. This is demonstrated by the fact that the longer the gap length, the greater the error (ME, MAE, and RMSE) and the lower the coefficient of determination of the regression between the real values and the gap filled values, as well as the relatively larger errors of the variables with a higher degree of variability in a short period of time (wind direction, for example, is the most unreliable to interpolate). The interpolated upward shortwave radiation series with three hours gaps differs

significantly ($p<0.05$) from the true values, for other variables evaluated, the difference is not significant. These findings suggest that the gap filling method used in this study can reasonably reconstruct the gaps within one to three hours.

Final QC code was assigned to each specific variable, indicating correct (0), suspicious (1), erroneous (2), gap filled (4), missing (8), and without any check performed (9).

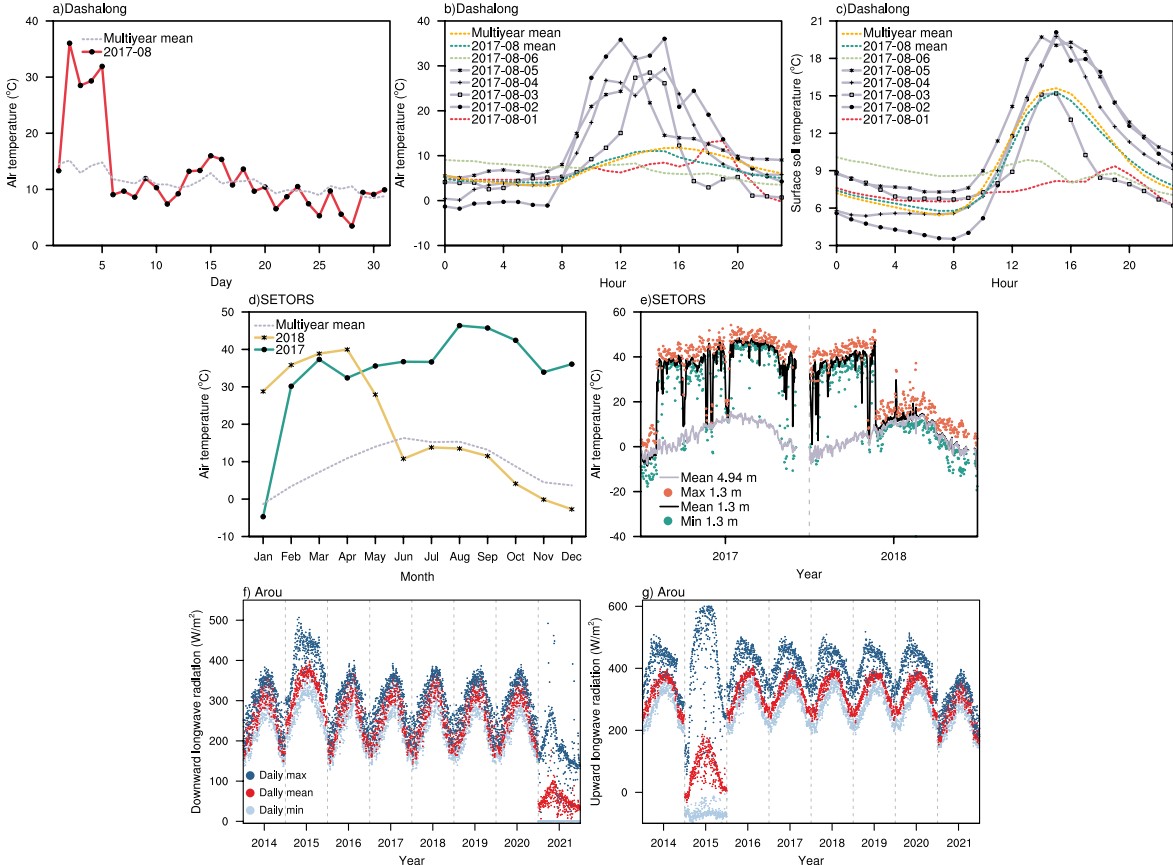

**Figure 3**. (a) Variation of daily mean air temperature in August 2017 and the multiyear average for the corresponding month at the Dashalong station, temperatures during the period of August 2-5, 2017 were within the acceptable bounds but significantly higher than neighbouring days, (b) diurnal variations of the air temperature observed during August 1-6, 2017, monthly mean diurnal cycle in August 2017, and the multiyear average for the August, (c) the diurnal variations of the surface soil temperature, (d) the seasonal variation of monthly mean 1.3 m height air temperature observed at 2017, 2018,

and its multiyear average at the SETORS station, the monthly mean air temperatures from February, 2017 to May, 2018

were significantly deviate from the multiyear mean, (e) the variations of daily minimal, mean and maximum air temperature observed at 1.3 m, and the daily mean temperatures observed at 4.94 m, (f) variation of daily minimal, mean and maximum downward longwave radiation and (g) upward longwave radiation observed at the Arou station.

### 2.3.4 Data archiving

During the archiving step, the header descriptions of the output files were first standardized to include information about the variable name, height/depth, and units. This information was expressed in the following format: variable_height/depth(units). The variable names are expressed as abbreviations which are listed in Appendix A. Two levels of data sets were produced for this study. Level 0 contains the hourly observations extracted from the raw data collection without any further processing applied and could contain errors and gaps in time. Level 1 is the quality-assured and partially

gap-filled observations.

## 3 Data description

### 3.1 Near-surface meteorological data

     To illustrate the near-surface micrometeorology characteristics of each station, Fig. 4 compares the diurnal variation of wind speed, air temperature, relative humidity, and air pressure at a monthly scale and the seasonal variation of daily mean

values. Note that the lowest layer was chosen for stations with gradient meteorological parameters observed, although observing height vary among stations, height adjustment is not implemented in this study. For the Yakou, Dashalong, and Jingyangling stations, the wind speeds were observed at a height of 10 m, which was much higher than other stations.

### 3.1.1 Wind speed

     Most stations exhibit discernible diurnal variation across the various seasons. Wind speed typically increased steadily from

noon onwards and peaked between 3 and 5 p.m. (depending on the station). This was followed by a gradual decrease (See Fig. 4a and c). Wind speed was generally steady and modest during the night, except for the Yakou station, with nighttime wind speeds there comparable to daytime wind speeds from October to December (Fig. 4a). Seasonally, winter wind speed observed at the TGL station exhibited the highest diurnal variation, whereas strong wind speeds throughout the day made the Yakou to experience higher mean wind speed on a daily scale than the others in October, November, December, and January. Winter is

often the season with the highest daily mean wind speed; however, this was not the case for the Arou, Jingyangling, and NASED stations, and winter also did not see the largest diurnal variation in these stations. The seasonal differences in wind speeds observed at Arou and SETORS were not significant. In comparison to other stations, the SETORS station had the lowest daily mean wind speeds, while its diurnal variance was more pronounced than that of the Arou station. Variations in wind speed between different geographic locations are caused by several crucial factors. One major influence is the

atmospheric pressure gradient forces, resulting from temperature gradients and variations in atmospheric pressure. This

pertains to the relatively higher wind speeds observed at TGL, QOMS, Ngoring Lake, and NAMORS stations. The wind in these regions accelerated due to thermal difference between the ice/lake and the land (Sun et al., 2007; Meng et al., 2023). Another factor to consider is the land surface conditions, including the roughness of the landscape that the air has to flow over and the local geographic features. Wind speed was typically higher in stations with smooth surfaces such as ice (TGL) or lakes (Ngoring Lake and NAMORS). While over rough surfaces, such as the SETORS, which is surrounded by forests that increase friction and slow winds down, the wind speed was relatively small. It is worth noting that wind speeds recorded at Dashalong station between May 2014 and September 2015 were notably lower than other periods.

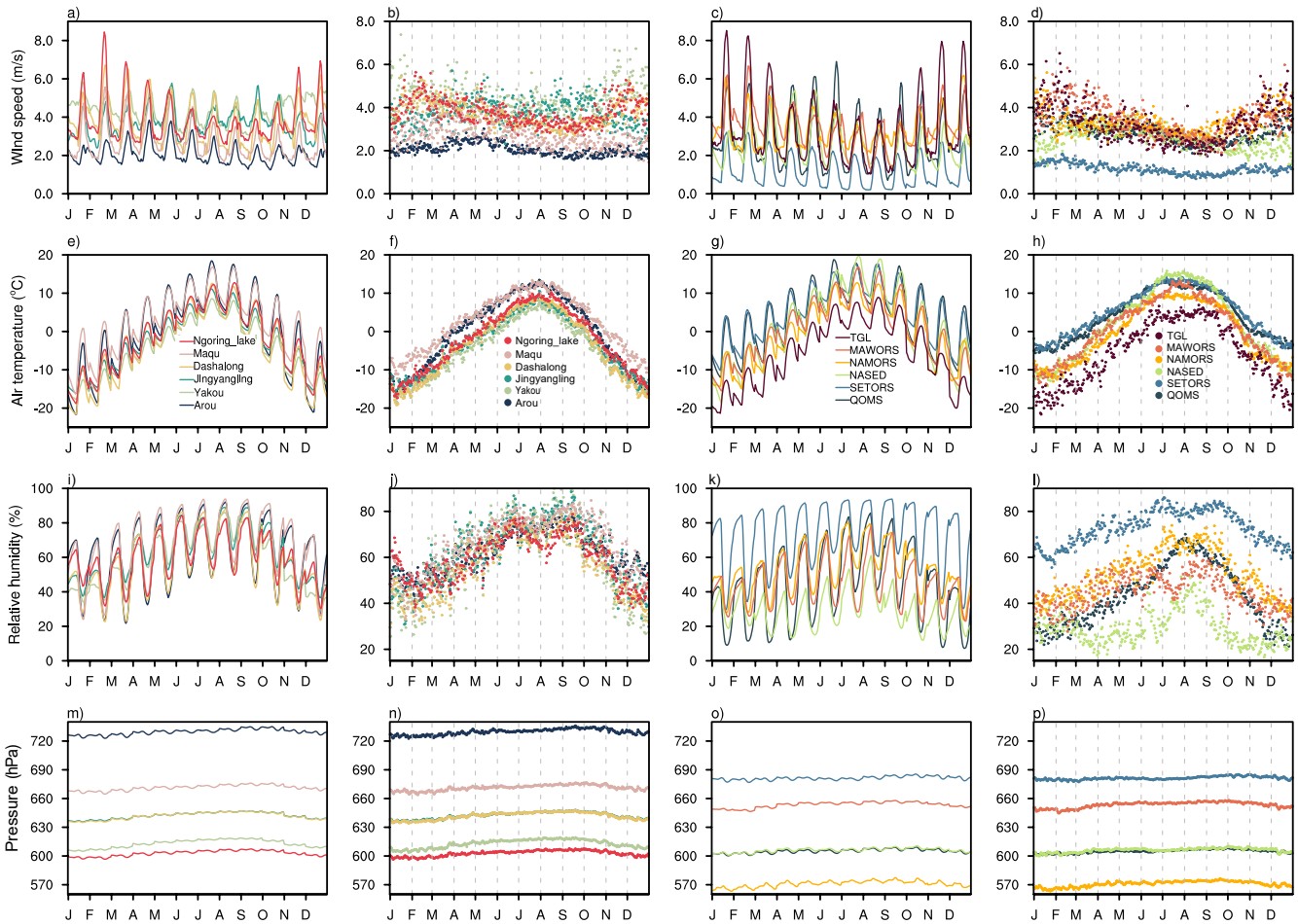

**Figure 4.** Seasonal variations of the diurnal (the first and the third column) and daily mean (the second and the fourth column) wind speed (a-d), air temperature (e-h), relative humidity (i-l), and air pressure (m-p) at the 12 stations.

### 3.1.2 Air temperature

The near-surface temperature did not follow a consistent declining trend with increasing altitude. In fact, air temperatures in certain lower elevation regions were even colder than sites with higher elevation. This difference was particularly noticeable

during the winter. For instance, the average daily mean winter temperature at Arou station (3,033 m) was about 5.4 °C lower compared to that at Maqu station (3,434 m), and 8.9 °C lower than that observed at QOMS station (4,276 m). The latitude difference among the stations is a key factor, as the maximum sunshine duration and the solar zenith angle are highly dependent on the latitude, influencing greatly the absorption of solar radiation. The seasonal variations in daily mean winter temperature from four stations (Arou, Dashalong, Jingyangling, and Yakou) located at similar latitudes confirmed this association (Fig. 4f). In this case, latitude is the determining factor in the rise of near-surface temperature at these three stations as increasing elevation. Additionally, elevation also has a significant impact on air temperature. For example, consider Maqu (3,434 m) and Ngoring Lake (4,280 m), which have comparable latitudes but range in elevation by around 800 m. The daily mean air temperature observed at the higher elevation station of Ngoring Lake was 4.5 °C colder than that of the Maqu station throughout the year (Fig. 4f). Moreover, the multiyear mean diurnal temperature variations indicate that the peak temperatures in Arou were noticeable higher than those recorded at the other three stations situated on the same latitude. The near-surface temperature at NASED (33.39°N, 4,270 m) was higher than that at the QOMS (28.36°N, 4,276 m) station (Fig. 4g and f). These two stations are located at roughly the same altitude with a latitude difference of nearly 5 degrees. The fact that more rainfall event occurs at the QOMS station during the monsoon season because of the summer monsoon may be the cause of this difference in air temperature.

### 3.1.3 Humidity

There was a negative correlation between the diurnal variation in relative humidity and air temperature when compared across stations. The peak relative humidity (Fig. 4i and k) typically matched with the coldest temperatures (Fig. 4e and g). This also coincided with the deep V-shaped variation in relative humidity observed at each station when the summer daily mean temperature reached its peak. Station-specific differences existed in the timing of the summertime reduction in relative humidity (Fig. 4j and l). The relative humidity varied greatly at both the diurnal and seasonal time scales for the QOMS and SETORS stations, which share comparable values and seasonal variance in the daily mean air temperature. It was observed that the daily mean relative humidity at the SETORS station remained consistently higher (above 60%) throughout the year. The wet atmosphere in SETORS station was closely associated with its location within the largest corridor for water vapor transport to the TP. In contrast, with relative humidity below 40% over most of the year save for the summer, NASED was the driest station except the TGL (where relative humidity observations are unavailable).

### 3.1.4 Air pressure

The diurnal and seasonal air pressures showed slight variation (Fig. 4m-p). Among the stations where air pressure observations were available (excluding the TGL station where air pressure was not recorded), the NAMORS station exhibited the lowest pressure due to its altitude being second only to that of the TGL station. Barometric pressure decreases with increasing altitude, with the highest recorded pressure being at the Arou station, which has the lowest altitude among all

stations. Moreover, although there were minor differences in altitude, stations like Dashalong and Jingyangling, QOMS, and

NASED had almost identical air pressure.

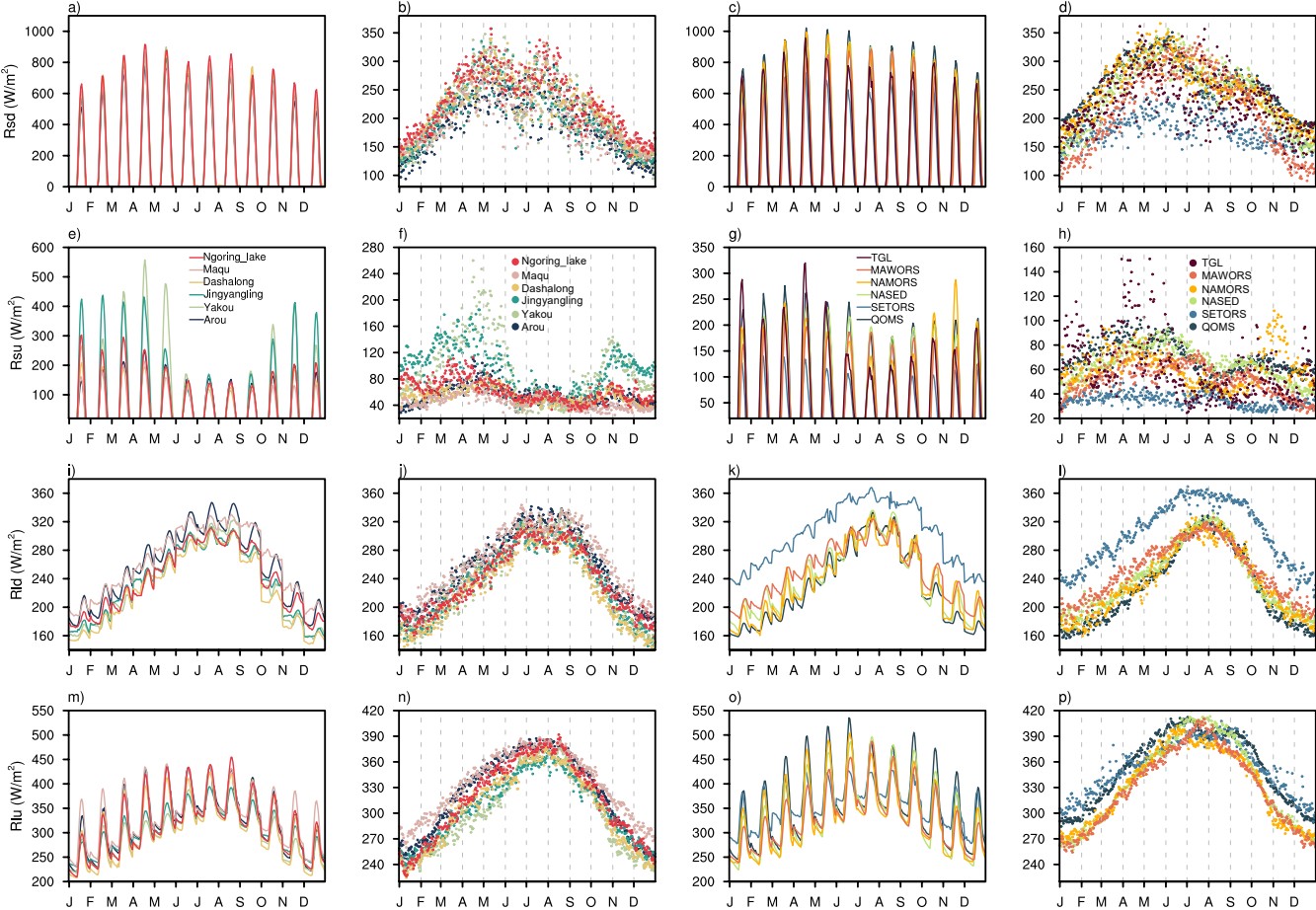

**Figure 5.** Seasonal variations of the diurnal (the first and the third column) and daily mean (the second and the fourth column) downward shortwave radiation (a-d), upward shortwave radiation (e-h), downward longwave radiation (i-l), and upward longwave radiation (m-p) at the 12 stations.

### 3.1.5 Radiations

The downward shortwave radiation (Rsd) demonstrated a similar pattern of variation across all stations during the summer. Initially, there was a decrease in Rsd and followed by a subsequent recovery (Fig. 5b and d). While specific stations (e.g., SETORS and TGL) started to decline in May, the majority of the decline in Rsd happened in June and July. Given that the majority of the precipitation on the TP occurs in summer, there may be a direct connection between this and the concurrent precipitation events. The seasonal variation in solar radiation over the TP was also significantly influenced by changes in cloud

cover. The upward shortwave radiation (Rsu) was very low during the rainy season (June to August), which can be explained

by the low surface albedo (Fig. 6) caused by factors like the damp soil and the heavy vegetation cover during this period. A rapid rise in Rsu was observed at Jingyangling, Yakou, TGL, Ngoring Lake, and NAMORS station in late fall and winter-spring, attributed to the high albedo of snow cover. Further analysis revealed that from early October through May of the following year, the surface albedo at Yakou and Jingyangling stations remained above 0.4 (Fig. 6). This result agreed with the changes in snow depth observed at the Yakou station (Che et al., 2019), indicating that snow plays a dominant role in the local hydrologic cycle. In contrast, Maqu and SETORS exhibit small Rsu, presumably owing to the greater vegetation coverage and little snow cover distributed in these areas.

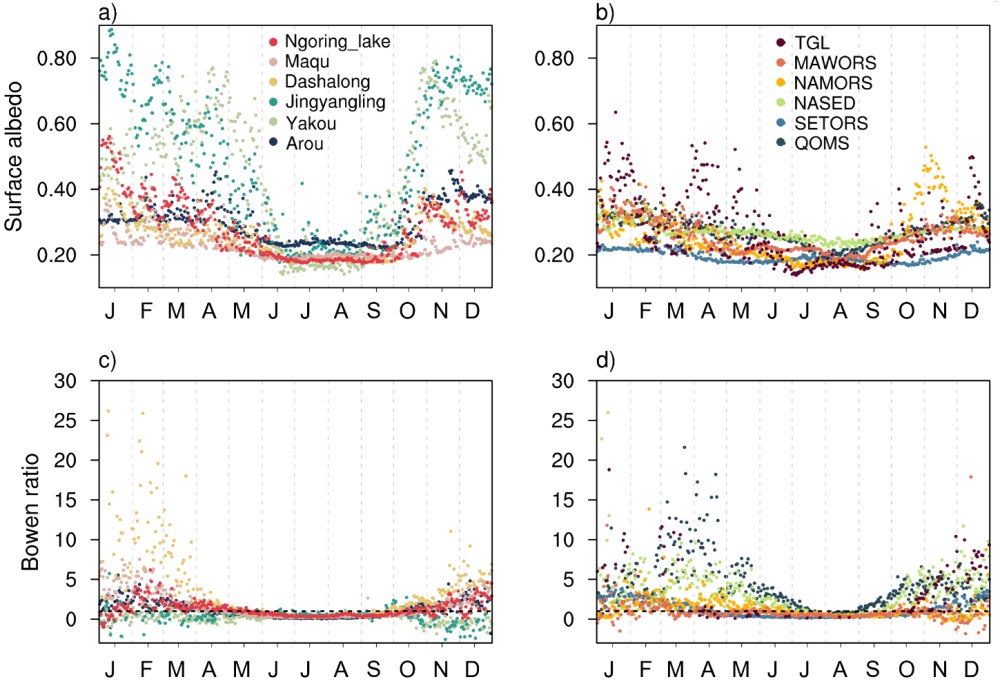

**Figure 6.** Seasonal variations of the surface albedo (a and b), and Bowen ratio (c and d).

There was a noticeable seasonal variation in downward longwave radiation (Rld), but the diurnal variability was not very large throughout the seasons. The highest values were recorded in summer (June to August) when the total cloud cover was high. The SETORS station showed consistently higher Rld compared to other stations throughout the year, which was attributable not only to the warmer ambient temperatures but also its cloud cover. A more thorough examination of the seasonal variations in near-surface temperature, relative humidity, and Rsd at the QOMS and SETORS station revealed that the difference in water vapor content of the air was also a significant contributor to the spatial variability of Rld across stations. The figure did not display both downward and upward longwave radiation from the TGL station since these observations failed the quality control test and were deemed anomalous. The QC codes of the Rld observed during the period from June 2016 to October 2019 were manually adjusted to 2 after the expert quality assessment (listed in Table 3).

**3.2 Soil hydrothermal data**

The soil hydrothermal process is crucial for the exchange of energy, moisture, and material between the land surface and the atmosphere. Fig. 7 compares the multi-year mean diurnal and seasonal variations of the shallow-layer (0.1 m depth) soil temperature and soil moisture to better illustrate the hydrothermal differences due to the spatial variability in soil physical and chemical properties (e.g., soil type, porosity, organic matter content), vegetation characteristics, and meteorological conditions between stations. The soil hydrothermal measurements observed at 0.2 m depth was chosen for comparison for the NASED

station, as observations at 0.1 m depth were not recorded prior to 2020. It should be noted that the gaps in the annual variation were caused by the existence of missing data on soil temperature and moisture, such as the poor continuity of soil hydrothermal observations at the TGL station.

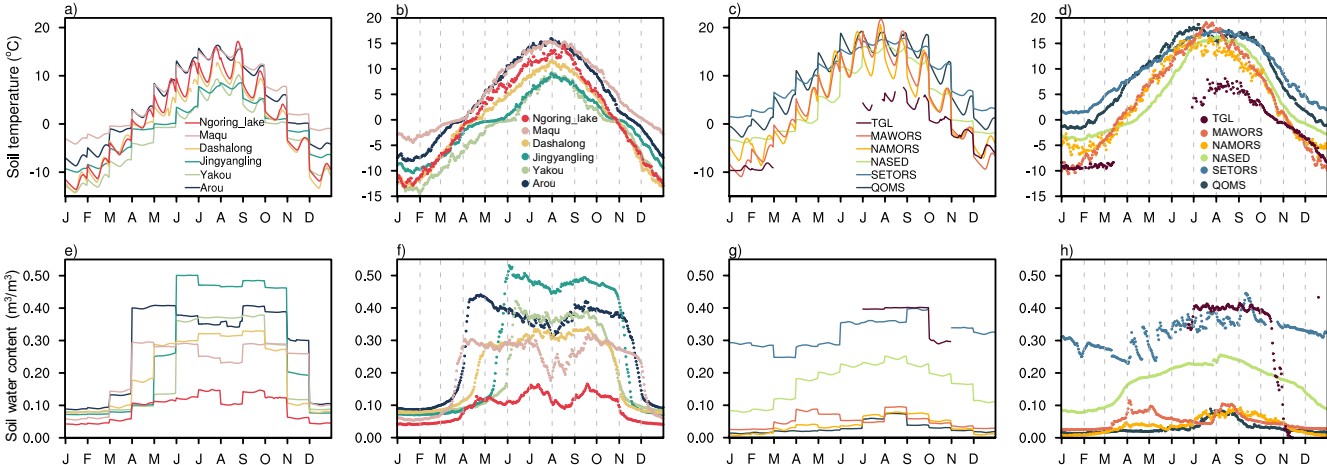

**Figure 7.** Seasonal variations of the diurnal (the first and the third column) and daily mean (the second and the fourth column) shallow-layer (0.1 m depth for stations except the NASED where 0.2 m was used) soil temperature (a-d), and soil

water content (e-h) at the 12 stations.

**3.2.1 Soil temperature**

    On a diurnal and seasonal basis, the soil temperature at 0.1 m depth at each station showed high consistency with the air temperature and upward longwave radiation. While the diurnal variations in soil temperature were not as great as those of the other two variables across the seasons, there were still appreciably greater spatial differences in soil temperature between

stations. The data quality control and interpolation of missing points in time series can both benefit from the consistent variations across different variables. At the SETORS station, the 0.1 m depth soil temperature stayed above 0 °C throughout the year, with only minor daily variations between seasons (Fig. 7c and d). Conversely, the daily mean soil temperature at the Yakou station can drop as low as -15 °C in winter. Specifically, it should be highlighted that the topsoil temperature (0 cm) is affected by multiple factors, resulting in a complex and substantial fluctuation with a high level of uncertainty. Therefore,

considerable care must be taken in selecting the topsoil temperature observations for analysis.

### 3.2.2 Soil moisture

Soil moisture showed two peaks throughout the year, with a reduction occurring primarily during the summer season, followed by an increase in early autumn. This tendency was more noticeable at the northeastern stations (Fig. 7f). Soil moisture increased rapidly with temperature exceeding 0 °C, and this was closely linked to the occurrence of precipitation and the resulting rise in surface runoff (Li et al., 2021). Summertime soil moisture was mostly dependent on precipitation, in the absence of precipitation, evapotranspiration and water infiltration caused a sharp decline in soil moisture at shallow depths. As shown in the surface albedo annual variation highlights that snowfall followed by snowmelt might contribute to the late summer and early fall rise in soil moisture. Because of the humid air, dense vegetation cover, and high soil temperature at the SETORS station, the shallow soil remained moist all year long. The 0.2 m soil moisture at the NASED station was much greater than that of the QOMS, NAMORS, and MAWORS stations, whose altitudes are equivalent to the NASED station. This suggests that the soil moisture profile had a large vertical variation.

### 3.3 Turbulent flux data

Sensible and latent heat fluxes are two main forms of heat transfer that occur between the surface and the atmosphere. Fig. 8 displays the diurnal variation as well as the annual fluctuation of these fluxes, with a QC flag labelled as 0 or 1. It should be specifically emphasized that the turbulent fluxes at the TGL station were seriously missing (see Fig. A3 for year-by-year data availability and data quality), hence, it is highly recommended to assess the data availability before analysis was performed.

At each station, the sensible heat flux ($H$) increased as temperature rose but declined when precipitation occurred and cloud cover increased. The month in which $H$ peaked varied greatly amongst stations; for instance, it typically peaked in March at SETORS and in June at QOMS. This was highly coincident with the beginning of precipitation events (Chen et al., 2012). The Yakou and Jingyangling stations did not experience a significant increase in $H$ in the spring due to the existence of surface snow cover. Instead, a rapid decrease of $H$ to negative values was observed in mid to late October, indicating that the surface absorbed energy in this period was primarily used for snowmelt. This emphasized the critical role of the seasonal snowpack on the surface energy budget. At night, there was a high negative $H$, suggesting the frequent occurrence of a significant temperature inversion in the near-surface layer. This result agrees with earlier studies (Chen et al., 2012). In contrast, positive nighttime turbulent fluxes were frequently observed at the Jingyangling station during May-October, but explaining the reasons behind this needs further analysis.

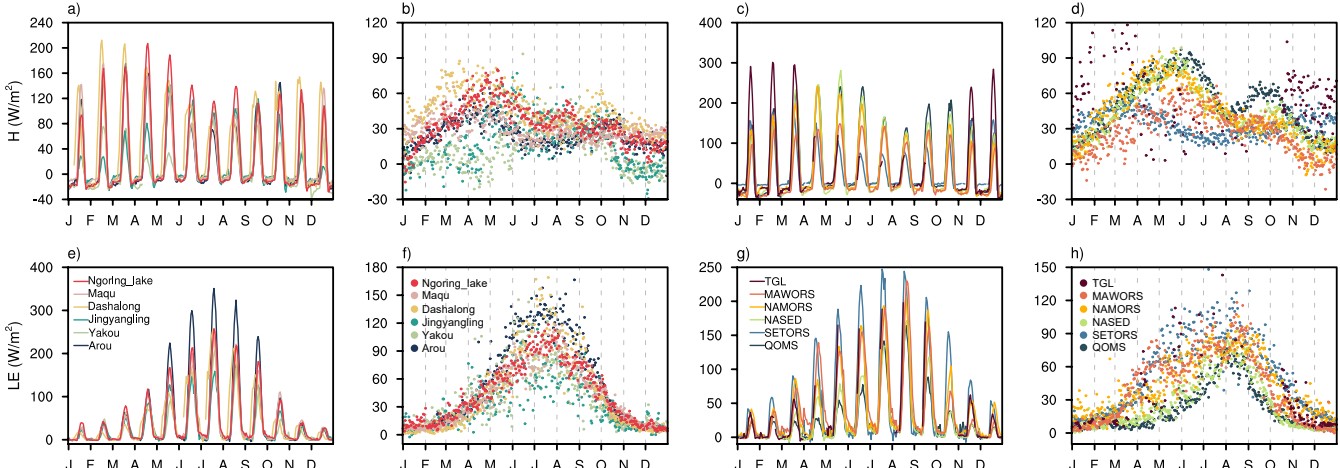

**Figure 8.** Seasonal variations of the diurnal (the first and the third column) and daily mean (the second and the fourth column) sensible heat flux (a-d), and latent heat flux (e-h) at the 12 stations.

With the gradual onset of precipitation at each station, the latent heat flux (*LE*) increased significantly both in terms of the diurnal variability and the daily average. During the rainy season, *LE* was highest at the Arou station on a diurnal scale while it was lowest at the Jingyangling station, with a maximum value not exceeding 200 W m$^{-2}$. From the seasonal variation of the Bowen ratio calculated based on the daily mean *H* and *LE* at each station, most of the Bowen ratios at Arou, Dashalong, Jingyangling, Yakou, Maqu, Ngoring Lake and NAMORS were below 1 from May to September, indicating the dominant role of *H* in the surface energy exchange in this period. From April to October, *H* dominated at the SETORS and MAWORS stations, while only July and August were dominated by *H* at the QOMS and NASED station. These results indicate that *H* was the primary atmospheric heat source in the TP during the spring and winter seasons.

## 3.4 Data consistency before and after the reconstruction of observation system

The observation system was adjusted in 2020 at ITP-affiliated stations, with minor differences in the observation heights/depths before and after the adjustment. We thoroughly checked the daily variations of each variable to a maintain consistent time series. Fig. 9 illustrates the interannual variation of daily mean micrometeorological variables observed at the QOMS station, where the first layer gradient measurement height remained unchanged. The data time series were in good consistency, indicating that different sensor types can accurately capture the atmospheric conditions, and that the adjustment had little impact on the observations. This is demonstrated by comparing the features of the seasonal variation in the minimum and maximum values, as well as the amplitude of the variations before and after the adjustment.

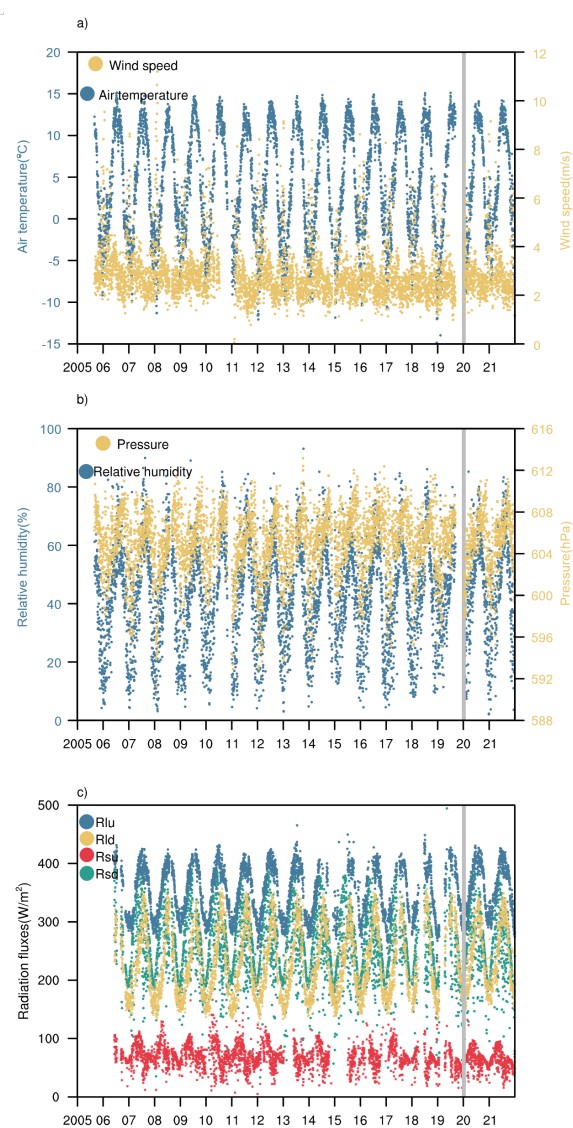


**Figure 9.** Variations in the wind speed (a), air temperature (a), air pressure (b), relative humidity (b), downward shortwave radiation (c), upward shortwave radiation (c), downward longwave radiation (c), and upward longwave radiation (c) observed at the QOMS station. The grey vertical line indicates the date the observation system was uprated.

## 4 Potential applications enabled from this integrated dataset

This integrated observation dataset for land-atmosphere interaction incorporates multiyear field observations from 12 stations that cover typical landscapes of the TP region and were established by multiple research organizations. Compared with prior datasets released (e.g., Ma et al., 2020; Meng et al., 2023), this dataset involves more stations with various surface characteristics, allowing for a more thorough characterization of the spatial heterogeneity of land-atmosphere interactions over

the TP. A strict data quality control procedure was implemented to process the dataset and quality flag was provided for each

record, this pioneering initiative facilitates data users access to reliable observations, and minimizes the use of erroneous data, enables its widespread usage in studying the earth system of the TP. Sustained land surface monitoring of the TP can provide pivotal constraints to advance Earth system modeling, remote sensing capabilities, and reanalysis accuracy for this climate-vulnerable region. More specifically, field observations conducted across various landscapes and scales are indispensable for gaining a comprehensive understanding of the interactions between the land surface and overlying atmospheres. Taking the

lake-atmosphere interactions as an example (e.g., Li et al., 2015; Wang et al., 2019), current field observations can provide fine-scale multi-component integrated observations at spatial and temporal scales ranging from centimeters to kilometers and from and from seconds to sub-hourly. By using the three-dimensional measurements from PBL tower, eddy covariance system, profile measurements of temperature, humidity, and wind by microwave radiometers, wind profiler, and radiosonde system, the physical mechanisms of land-atmosphere and boundary layer processes over the TP can be systematically investigated.

Furthermore, field measurements are widely used to derive or calibrate land surface parameters for regional-scale estimate, satellite retrieval, and numerical simulation of energy and water exchange over heterogeneous landscape (Yang et al., 2008; Chen et al., 2013). There is no doubt that this enhanced observation network enables a systematic assessment of model robustness and uncertainty in representing the land-atmosphere interactions in complex mountainous regions, providing better guidance for physical parameterization optimization of numerical models involving cryospheric, hydrologic, and atmospheric

processes in the intricate TP terrain. Meanwhile, extensive field measurements are critical for validating, calibrating, and refining of remote sensing retrieval algorithms over the topographically complicated terrain. For instance, Yuan et al., (2021) used in-situ measurements from this dataset to present an enhanced canopy transpiration model as well as an improved approach for calculating soil evaporation using soil moisture and texture. Systemic biases in key land surface parameters in the reanalysis products can be decreased by integrating synthesis ground-based datasets and revised satellite products through

sophisticated data assimilation techniques. For instance, Qi et al., (2023) improved the accuracy of land surface temperature retrieval over the TP based on the in-situ data. The combination of credible datasets provides multidimensional insights into the intricate mechanisms driving the recent changes across the fragile environments of the TP. This makes possible to comprehend the TP's critical role in Asian monsoons, water resources, and global climate teleconnections. In addition, predicting future changes and developing adaptive strategies for the environment and communities of the TP that are currently

experiencing disproportionate climate change impacts depend on these integrated land surface observations.

**5 Data availability**

The dataset of near-surface micrometeorology, soil hydrothermal, and turbulent fluxes observations can be open-accessed from the National Tibetan Plateau Data Center (https://doi.org/10.11888/Atmos.tpdc.300977, Ma et al., 2023a). It is available in two distinct directories that include raw data on an hourly scale (level 0) and quality-controlled data (level 1). Each

observation station is assigned a subfolder within the directories. The data was divided into three categories: gradient

meteorological data (Met), soil hydrothermal data (Soil), and turbulent flux data (Flux). The CSV-formatted output files, which follow the naming convention datatype_station_year.csv (level 0) and QC_datatype_station_year.csv (level 1), are accessible for download. The data header of each CSV-formatted data file contains comprehensive information on the variable units and heights/depths, with naming format: variable_height/depth (units). The variable names are expressed as abbreviations. Appendix A provides a full list of the abbreviations for each variable. UTC+8 was adopted as the time standard for all the data files. Furthermore, a separate file provides information about data integrity and the percentage of correct data on a monthly scale. This information is presented yearly in Appendix B.

**Conclusions**

This paper presents a suite of integrated field observations of land-atmosphere interactions over the TP with the cooperation of several agencies and organizations dedicated to field observations throughout the TP over several decades. This dataset includes hourly measurements of soil hydrothermal properties, near-surface micrometeorological conditions from 12 stations spanning up to 17 years (2005-2021). This paper highlights the complexity and spatial heterogeneity of land-atmosphere interactions over the mountainous region by describing in detail the observation network and presenting the hydrometeorological characteristics, soil hydrothermal properties, and surface energy balance components of these stations covering various landscapes over the decades. All of the data series in this dataset have been quality controlled using a combination of automatic error screening, manual inspection, diagnostic checking, adjustments, and quality flagging, as compared to other similar datasets that have previously been released. Suspicious and erroneous data were identified, and a QC code was assigned to each variable value. The specially designed data processing procedures tailored to handle the data issues of this integrated network is detailed described. It is indisputable that the long-term hourly quality-assured dataset presented here will contribute to a broad research effort and help advance the fine-scale understanding of the land-atmosphere interactions over the heterogeneous TP region, refine land surface models, reanalysis products, and remote sensing retrievals.

**Author contribution.** YMM, XL, and YYC developed and shaped the overall workflow for compiling this integrated dataset, and review & editing the manuscript. ZPX developed the automated QC algorithm, prepared the data, and led the writing of this article. SML, TC, ZWX, LYS, XBH, WQM, XHM, SML, BQX, HBZ, JBW and GJW provided the observations and review & editing the manuscript.

**Competing interests.** The authors declare that they have no conflict of interest.

**Acknowledgements.** We would like to thank all the scientists, engineers and students who participated in the field observations, instruments maintenance, and data processing. We highly appreciate valuable and constructive comments on the manuscript

provided by the anonymous reviewers. Computing resources for data processing was supported by the National Key Scientific and Technological Infrastructure project "Earth System Numerical Simulation Facility" (EarthLab).

**Financial support.** This research has been supported by the Second Tibetan Plateau Scientific Expedition and Research (STEP) program (2019QZKK0103, 2019QZKK020109), the National Natural Science Foundation of China (42230610, 42375075, U2242208), and Tibet Science and Technology Program (XZ202301ZY0001G).

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

**Table 1. Overview of the sensors used at each station.**

| Site | Variables | Sensors models | Manufacturers | Period | Heights | Units |
|---|---|---|---|---|---|---|
| QOMS<br>Lat: 28.36 ºN<br>Lon: 86.95 ºE<br>Alt: 4,276 m<br>Alpine desert | Air temperature | HMP45C-GM | Vaisala | 2005-2019 | 1.5, 2.0, 4.0, 10.0, and 20.0 m | ºC |
| | | HMP155A-L | Vaisala | 2019-present | 1.5, 2.5, 4.0, 10.0, and 20.0 m | |
| | Wind speed and direction | 034B | MetOne | 2005-2019 | 1.5, 2.0, 4.0, 10.0, and 20.0 m | m s⁻¹, º |
| | | 05103-L | R.M.Young | 2019-present | 1.5, 2.5, 4.0, 10.0, and 20.0 m | |
| | Humidity | HMP45C-GM | Vaisala | 2005-2019 | 1.5, 2.0, 4.0, 10.0, and 20.0 m | % |
| | | HMP155A-L | Vaisala | 2019-present | 1.5, 2.5, 4.0, 10.0, and 20.0 m | |
| | Pressure | PTB220A | Vaisala | 2005-2019 | - | hPa |
| | | CS106 | Vaisala | 2019-present | | |
| | Radiations | CNR1 | Kipp & Zonen | 2005-2019 | 1.5 m | W m⁻² |
| | | CNR4 | Kipp & Zonen | 2019-present | | |
| | Precipitation | RG13H | Vaisala | 2005-2019 | 1.0 m | mm |
| | | RG3-M | Onset | 2019-present | | |
| | Soil temperature | Model 107 | Campbell | 2005-2019 | 0.1, 0.2, 0.4, 0.8 and 1.6 m | ºC |
| | | CS655 | Campbell | 2019-present | | |
| | Soil moisture | CS616 | Campbell | 2005-2019 | 0.1, 0.2, 0.4, 0.8 and 1.6 m | m³ m⁻³ |
| | | CS655 | Campbell | 2019-present | | |
| | Soil heat flux | HFP01 | Hukseflux | 2005-2019 | 0.05 m | W m⁻² |
| | | HFP01 | Hukseflux | 2019-present | 0.1 and 0.2 m | |
| | EC | CSAT3 | Campbell | 2007-2019 | 3.25 m | |
| | | LI-7500 | Li-COR | | | |
| | | CSAT3B | Campbell | 2019-present | 3.0 m | |
| | | LI-7500DS | Li-COR | | | |
| SETORS<br>Lat: 29.77 ºN<br>Lon: 94.73 ºE<br>Alt: 3,327 m<br>Alpine meadow | Air temperature | HMP45C-GM | Vaisala | 2007-2020 | 1.3, 4.94, 9.95, and 18 m | ºC |
| | | HMP155A-L | Vaisala | 2020-present | 1.5, 2.5, 4.0, 10.0 and 20.0 m | |
| | Wind speed and direction | 034B | MetOne | 2007-2020 | 1.3, 4.94, 9.95, and 18 m | m s⁻¹, º |
| | | 05103-L | R.M.Young | 2020-present | 1.5, 2.5, 4.0, 10.0 and 20.0 m | |
| | Humidity | HMP45C-GM | Vaisala | 2007-2020 | 1.3, 4.94, 9.95, and 18 m | % |
| | | HMP155A | Vaisala | 2020-present | 1.5, 2.5, 4.0, 10.0 and 20.0 m | |
| | Pressure | PTB220A | Vaisala | 2007-2020 | - | hPa |
| | | PTB110 | Vaisala | 2020-present | | |
| | Radiations | CNR1 | Kipp & Zonen | 2007-2020 | - | W m⁻² |
| | | CNR4 | Kipp & Zonen | 2020-present | | |
| | Precipitation | RG13H | Vaisala | 2007-2020 | - | mm |
| | | RG3-M | Onset | 2020-present | | |
| | Soil temperature | Model 107 | Campbell | 2007-2020 | 0.04, 0.1, 0.2, 0.6 and 1 m | ºC |
| | | CS655 | Campbell | 2020-present | 0.1, 0.2, 0.4, 0.8, 1.6 m | |
| | Soil moisture | CS616 | Campbell | 2007-2020 | 0.04, 0.1, 0.2, 0.6 and 1 m | m³ m⁻³ |
| | | CS655 | Campbell | 2020-present | 0.1, 0.2, 0.4, 0.8, 1.6 m | |

| | | | | | | |
|---|---|---|---|---|---|---|
| | Soil heat flux | HFP01 | Hukseflux | 2007-2020 | 0.04, 0.1, 0.2, 0.6 and 1 m | W m⁻² |
| | | HFP01SC | Hukseflux | 2020-present | 0.1 and 0.2 m | |
| | EC | CSAT3 | Campbell | 2007-2020 | 3.04 m | |
| | | LI-7500 | Li-COR | | | |
| | | CSAT3B | Campbell | 2020-present | 3.13 m | |
| | | LI-7500DS | Li-COR | | | |
| NASED Lat: 33.39 ºN Lon: 79.70 ºE Alt: 4,270 m Alpine desert | Air temperature | HMP45C | Vaisala | 2009-2020 | 1.5 and 2.8 m | ºC |
| | | HMP155A-L | Vaisala | 2020-present | 1.0, 2.0, 4.0, 10.0 and 20.0 m | |
| | Wind speed and direction | 05013 | RM Young | 2009-2020 | 1.5 m | m s⁻¹, º |
| | | 05103-L | R.M.Young | 2020-present | 1.0, 2.0, 4.0, 10.0 and 20.0 m | |
| | Relative humidity | HMP45C | Campbell | 2009-2020 | 1.5 and 2.8 m | % |
| | | HMP155A | Vaisala | 2020-present | 1.0, 2.0, 4.0, 10.0 and 20.0 m | |
| | Pressure | PTB210 | Vaisala | 2009-2020 | - | hPa |
| | | PTB110 | Vaisala | 2020-present | | |
| | Radiations | NR01 | Hukseflux | 2009-2020 | - | W m⁻² |
| | | CNR4 | Kipp & Zonen | 2020-present | | |
| | Precipitation | T-200B | Geonor | 2009-2020 | - | mm |
| | | RG3-M | Onset | 2020-present | | |
| | Soil temperature | CSI 109 | Campbell | 2011-2020 | 0.2, 0.5, 1.0 and 2.0 m | ºC |
| | | CS655 | Campbell | 2020-present | 0.1, 0.2, 0.4, 0.8, 1.6 m | |
| | Soil moisture | CS616 | Campbell | 2011-2020 | 0.2, 0.5, 1.0 and 2.0 m | m³ m⁻³ |
| | | CS655 | Campbell | 2020-present | 0.1, 0.2, 0.4, 0.8, 1.6 m | |
| | Soil heat flux | HFP01SC | Hukseflux | 2020-present | 0.1 and 0.2 m | W m⁻² |
| | EC | CSAT3 | Campbell | 2005-2020 | 2.75 m | |
| | | LI-7500 | Li-COR | | | |
| | | CSAT3B LI-7500DS | Campbell Li-COR | 2020-present | 3.7 m | |
| MAWORS Lat: 38.41ºN Lon: 75.05 ºE Alt: 3,647 m Alpine meadow | Air temperature | HMP155A | Vaisala | 2010-2020 | 1.9 m | ºC |
| | | HMP155A | Vaisala | 2020-present | 1.5, 3.0, 5.0, 10.0 and 20.0 m | |
| | Wind speed and direction | 05103-L | RM Young | 2010-2020 | 2 m | m s⁻¹, º |
| | | 05103 | R.M.Young | 2020-present | 1.5, 3.0, 5.0, 10.0 and 20.0 m | |
| | Relative humidity | HMP155A | Vaisala | 2010-2020 | 1.9 m | % |
| | | HMP155A | Vaisala | 2020-present | 1.5, 3.0, 5.0, 10.0 and 20.0 m | |
| | Precipitation | RG3-M | Onset | 2020-present | - | mm |
| | Pressure | PTB210 | Vaisala | 2010-2020 | - | hPa |
| | | PTB110 | Vaisala | 2020-present | | |
| | Radiations | NR01 | Hukseflux | 2010-2020 | - | W m⁻² |
| | | CNR4 | Kipp & Zonen | 2020-present | | |
| | Soil temperature | CSI 109 | Campbell | 2010-2020 | 0.1, 0.2, 0.4, 0.8 and 1.60 m | ºC |
| | | CS655 | Campbell | 2020-present | 0.1, 0.2, 0.4, 0.8 and 1.6 m | |
| | Soil moisture | CS616 | Campbell | 2010-2020 | 0.1, 0.2, 0.4, 0.8 and 1.60 m | v/v |

| | | CS655 | Campbell | 2020-present | 0.1, 0.2, 0.4, 0.8 and 1.6 m | |
|---|---|---|---|---|---|---|
| | Soil heat flux | HFP01SC | Hukseflux | 2020-present | 0.1 and 0.2 m | |
| | EC | CSAT3 LI-7500 | Campbell Li-COR | 2007-2020 | 2.3 m | |
| | | CSAT3B LI-7500DS | Campbell Li-COR | 2020-present | 3 m | |
| NAMORS Lat: 30.77 ºN Lon: 90.98 ºE Alt: 4,730 m Alpine steppe | Air temperature | HMP45D | Vaisala | 2005-2019 | 1.5, 2.0, 4.0, 10.0 and 20.0 m | ºC |
| | | HMP155A | Vaisala | 2020-present | 1.5, 2.5, 4.0, 10.0 and 20.0 m | |
| | Wind speed | WAA151 | Vaisala | 2005-2019 | 1.5, 10.0 and 20.0 m | m s$^{-1}$ |
| | | 05103 | R.M.Young | 2020-present | 1.5, 2.5, 4.0, 10.0 and 20.0 m | |
| | Wind direction | 020C | MetOne | 2005-2019 | 1.5, 10.0 and 20.0 m | º |
| | | 05103 | R.M.Young | 2020-present | 1.5, 2.5, 4.0, 10.0 and 20.0 m | |
| | Humidity | HMP45D | Vaisala | 2005-2019 | 1.5, 2.0, 4.0, 10.0 and 20.0 m | % |
| | | HMP155A | Vaisala | 2020-present | 1.5, 2.5, 4.0, 10.0 and 20.0 m | |
| | Pressure | PTB210 | Vaisala | 2005-2019 | - | hPa |
| | | PTB110 | Vaisala | 2020-present | | |
| | Radiations | CMP6 | Vaisala | 2005-2019 | - | W m$^{-2}$ |
| | | CNR4 | Kipp & Zonen | 2020-present | | |
| | Precipitation | RG13H | Vaisala | 2005-2019 | - | mm |
| | | RG3 | Onset | 2020-present | | |
| | Soil temperature | Model 107 | Campbell | 2005-2019 | 0, 0.1, 0.2, 0.4, 0.8 and 1.6 m | ºC |
| | | CS655 | Campbell | 2020-present | 0.1, 0.2, 0.4, 0.8 and 1.6 m | |
| | Soil moisture | CS616 | Campbell | 2005-2019 | 0, 0.1, 0.2, 0.4, 0.8 and 1.6 m | m$^3$ m$^{-3}$ |
| | | CS655 | Campbell | 2020-present | 0.1, 0.2, 0.4, 0.8 and 1.6 m | m$^3$ m$^{-3}$ |
| | Soil heat flux | HFP01SC | Hukseflux | 2020-present | 0.1 and 0.2 m | |
| | EC | CSAT3 LI-7500 | Campbell Li-COR | 2005-2019 | 3.06m | |
| | | CSAT3B LI-7500DS | Campbell Li-COR | 2020-present | 3.77 m | |
| TGL Lat: 33.03 ºN Lon: 92.007 ºE Alt: 5,150m Alpine Swamp meadow | Air temperature | 109 | Campbell | 2018-2021 | 1.5 m | ºC |
| | Wind speed | 0513 | R.M.Young | 2018-2021 | 1.5 m | m s$^{-1}$ |
| | Wind direction | 0513 | R.M.Young | 2018-2021 | 1.5 m | º |
| | Radiations | NR01 | Hukseflux | 2018-2021 | 1.5 m | W m$^{-2}$ |
| | Soil temperature | Hydra | Stevens | 2018-2021 | 0.1, 0.2, 0.3, 0.4, 0.5, 0.7, 0.9, and 1.1 m | ºC |
| | Soil moisture | Hydra | Stevens | 2018-2021 | 0.1, 0.2, 0.3, 0.4, 0.5, 0.7, 0.9, and 1.1 m | m$^3$ m$^{-3}$ |
| | EC | CSAT3 LI-7500 | Campbell Li-COR | 2019-2021 | 2.5 m | |
| Ngoring Lake Lat: 34.91 ºN Lon: 97.55 ºE | Air temperature | HMP45C | Vaisala | 2012-2019 | 3.2 m | ºC |
| | Wind speed and direction | CSAT3 | Campbell | 2012-2019 | 3.2 m | m s$^{-1}$, º |

| | | | | | | |
|---|---|---|---|---|---|---|
| Alt: 4,280 m | Humidity | HMP45C | Vaisala | 2012-2019 | 3.2 m | g kg$^{-1}$ |
| Alpine steppe | Pressure | PTB110 | Vaisala | 2012-2019 | - | kPa |
| | Radiations | CNR1 | Kipp & Zonen | 2012-2014 | 1.5 m | W m$^{-2}$ |
| | | CNR4 | Kipp & Zonen | 2014-2019 | | |
| | Precipitation | RG3-M | Onset | 2012-2012 | 1.3 m | mm |
| | | T200B | Geonor | 2013-2019 | - | |
| | Soil temperature | 109SS | Campbell | 2012-2019 | 0.05, 0.1, 0.2 and 0.4 m | ºC |
| | Soil moisture | CS616 | Campbell | 2012-2019 | 0.05, 0.1, 0.2 and 0.4 m | m$^3$ m$^{-3}$ |
| | Soil heat flux | HFP01 | Hukseflux | 2012-2019 | 0.05 m | W m$^{-2}$ |
| | EC | CSAT3 LI-7500 | Campbell Li-COR | 2012-2019 | 3.2 m | |
| Maqu | Air temperature | HMP45C | Vaisala | 2014-2019 | 2.75 m | ºC |
| Lat: 33.92 ºN | Wind speed and direction | CSAT3 | Campbell | 2014-2019 | 2.75 m | m s$^{-1}$, º |
| Lon: 102.15 ºE | Humidity | HMP45C | Vaisala | 2014-2019 | 2.75 m | g kg$^{-1}$ |
| Alt: 3,434 m | Pressure | PTB110 | Vaisala | 2014-2019 | - | kPa |
| Alpine steppe | Radiations | CNR1 | Kipp&Zonen | 2014-2019 | 1.5 m | W m$^{-2}$ |
| | Precipitation | T200B | Geonor | 2014-2019 | 1.5 m | mm |
| | Soil temperature | 109SS | Campbell | 2014-2019 | 0.05, 0.1, 0.2 and 0.4 m | ºC |
| | Soil moisture | CS616 | Campbell | 2014-2019 | 0.05, 0.1, 0.2 and 0.4 m | m$^3$ m$^{-3}$ |
| | Soil heat flux | HFP01 | Hukseflux | 2014-2019 | 0.08 m | W m$^{-2}$ |
| | EC | CSAT3 EC150 | Campbell | 2014-2019 | 2.75 m | |
| Arou | Air temperature | HMP45C | Vaisala | 2014-2021 | 1.0, 2.0, 5.0, 10, 15 and 25 m | ºC |
| Lat: 38.047 ºN | Wind speed | 010C | MetOne | 2014-2021 | 1.0, 2.0, 5.0, 10, 15 and 25 m | m s$^{-1}$ |
| Lon: 100.464 ºE | Wind direction | 020C | MetOne | 2014-2021 | 10 m | º |
| | Humidity | HMP45C | Vaisala | 2014-2021 | 1.0, 2.0, 5.0, 10, 15 and 25 m | % |
| Alt: 3,033 m | Pressure | PTB110 | Vaisala | 2014-2021 | 2 m | hPa |
| Alpine grassland | Radiations | CNR4 | Kipp&Zonen | 2014-2021 | 5 m | W m$^{-2}$ |
| | Precipitation | TE525M | Texas Electronics | 2014-2021 | 28 m | mm |
| | Soil temperature | CSI 109 | Campbell | 2014-2021 | 0, 2, 4, 6, 10, 15, 20, 30, 40, 60, 80, 120, 160, 200, 240, 280 and 320 cm | ºC |
| | Soil moisture | CS616 | Campbell | 2014-2021 | 2, 4, 6, 10, 15, 20, 30, 40, 60, 80, 120, 160, 200, 240, 280 and 320 cm | m$^3$ m$^{-3}$ |
| | EC | CSAT3 LI-7500A | Campbell Li-COR | 2014-2021 | 3.5 m | |
| Yakou | Air temperature | HMP45C | Vaisala | 2015-2021 | 5.0 m | ºC |
| Lat: 38.014 ºN | Wind speed | 010C | MetOne | 2015-2021 | 10 m | m s$^{-1}$ |
| Lon: 100.242ºE | Wind direction | 020C | MetOne | 2015-2021 | 10 m | º |
| | Humidity | HMP45C | Vaisala | 2015-2021 | 5.0 m | % |

| | | | | | | |
|---|---|---|---|---|---|---|
| Alt: 4,148 m Alpine meadow | Pressure | PTB110 | Vaisala | 2015-2021 | 2.0 m | hPa |
| | Radiations | CNR1 | Kipp&Zonen | 2015-2021 | 6.0 m | W m$^{-2}$ |
| | Precipitation | TE525M | Texas Electronics | 2015-2021 | 2.0 m | mm |
| | Soil temperature | CSI 109 | Campbell | 2015-2021 | 0, 4, 10, 20, 40, 80, 120, and 160 cm | ºC |
| | Soil moisture | CS616 | Campbell | 2015-2021 | 4, 10, 20, 40, 80, 120, and 160 cm | m$^3$ m$^{-3}$ |
| | EC | CSAT3 LI-7500A | Campbell Li-COR | 2015-2021 | 3.2 m | |
| Jingyangling Lat: 37.838 ºN Lon: 101.116ºE Alt: 3,750 m Alpine meadow | Air temperature | HMP45C | Vaisala | 2014-2021 | 5.0 m | ºC |
| | Wind speed | Windsonic | Gill | 2014-2021 | 10 m | m s$^{-1}$ |
| | Wind direction | Windsonic | Gill | 2014-2021 | 10 m | º |
| | Humidity | HMP45C | Vaisala | 2014-2021 | 5.0 m | % |
| | Pressure | PTB110 | Vaisala | 2014-2021 | 2.0 m | hPa |
| | Radiations | CNR4 | Kipp&Zonen | 2014-2021 | 6.0 m | W m$^{-2}$ |
| | Precipitation | TE525M | Texas Electronics | 2014-2021 | 10 m | mm |
| | Soil temperature | CSI 109 | Campbell | 2014-2021 | 0, 4, 10, 20, 40, 80, 120, and 160 cm | ºC |
| | Soil moisture | CS616 | Campbell | 2014-2021 | 4, 10, 20, 40, 80, 120, and 160 cm | m$^3$ m$^{-3}$ |
| | EC | CSAT3 LI-7500A | Campbell Li-COR | 2018-2021 | 4.5 m | |
| Dashalong Lat: 38.840 ºN Lon: 98.941ºE Alt: 3739 m Alpine meadow | Air temperature | HMP45C | Vaisala | 2014-2021 | 5.0 m | ºC |
| | Wind speed | 010C | MetOne | 2014-2021 | 10 m | m s$^{-1}$ |
| | Wind direction | 020C | MetOne | 2014-2021 | 10 m | º |
| | Humidity | HMP45C | Vaisala | 2014-2021 | 5.0 m | % |
| | Pressure | CS100 | Setra | 2014-2021 | - | hPa |
| | Radiations | CNR1 | Kipp&Zonen | 2014-2021 | 6.0 m | W m$^{-2}$ |
| | Precipitation | TE525M | Texas Electronics | 2014-2021 | 10 m | mm |
| | Soil temperature | CSI 109 | Campbell | 2014-2021 | 0, 4, 10, 20, 40, 80, 120, and 160 cm | ºC |
| | Soil moisture | CS616 | Campbell | 2014-2021 | 4, 10, 20, 40, 80, 120, and 160 cm | m$^3$ m$^{-3}$ |
| | EC | CSAT3 LI-7500RS | Campbell Li-COR | 2014-2021 | 4.5 m | |


**Table 2. Constant limits used for the range checks and temporal checks.**

| Variable | Unit | Min | Max | Change gradient ($\Delta x\ h^{-1}$) |
|---|---|---|---|---|
| Wind speed | m s$^{-1}$ | 0 | 30 | 0.5 |
| Wind direction | Degree | 0 | 360 | Not applicable[1] |
| Air temperature | °C | -35 | 40 | 0.1 |
| Relative humidity | % | 1 | 100 | 1 |
| Air pressure | hPa | 550 | 750 | 0.1 |
| Soil temperature | °C | -40…-10[2] | 15…40 | Not applicable[1] |
| Soil moisture | m$^3$ m$^{-3}$ | 0 | 1 | Not applicable[1] |
| Downward shortwave radiation | W m$^{-2}$ | Not applicable[1] | 1368 | Not applicable[1] |
| Upward shortwave radiation | W m$^{-2}$ | 0 | 1368 | Not applicable[1] |
| Downward longwave radiation | W m$^{-2}$ | 100 | 600 | Not applicable[1] |
| Upward longwave radiation | W m$^{-2}$ | 100 | 600 | Not applicable[1] |

1. Check is not implemented.

2. A range of values is given because they are defined individually for each site.

**Table 3. A full list of manually adjustments on QC score after expert quality assessment**

| Site | Target variable | Height/Depth | Period | Description of the issue | Treatment |
|---|---|---|---|---|---|
| Arou | Rld | - | 2015, 2021 | Most of the incorrect data in Rld in 2015 can be accurately detected by the automated system, however, data points that were within the threshold range but significantly higher in other years cannot be diagnosed | The QC codes of the significant high data were manually adjusted to 1 |
| | Rlu | - | 2015, 2021 | The issue was mostly consistent with the Rld data, yet some of the data points were noticeably lower than in other years | The QC codes of the significant high data were manually adjusted to 1 |
| | Rsd | - | 2015, 2021 | There was no seasonal variation in the data, and all values were noticeably lower than in other years | Label all data as errors |
| | Rld | - | 2015, 2021 | There was no seasonal variation in the data, and all values were noticeably higher than in other years | Label all data as errors |
| | SM | 3.2 m | 2015, 2021 | The seasonal variation of the data during this period was noticeably different from that of other years | Label all data as errors |
| Jingyangling | Rlu | - | 2016-2021 | A few successive data points had values within the threshold range but their seasonal variation was abnormal | Label all data as errors |
| | Ta | 5 m | 2017.07.17-2017.08.05 | Temperatures during this period were far higher than those of other years over the | Label all data as errors |

| | | | | | |
|---|---|---|---|---|---|
| | | | | same period, sometimes exceeding 20 °C, which has never been recorded. | |
| | WS | 10 m | 2014.10.29-2017.07.15 | The seasonal variation of wind speed in terms of daily mean, minimum, and maximum during this period were noticeably lower than in other periods | The QC codes were manually adjusted to 2 |
| | H and LE | - | 2021.01.01-2021.12.31 | Abnormally large turbulent fluxes were frequently observed | The QC codes were manually adjusted to 2 |
| NASED | H | - | 2019.05.14 | A very large negative value (-410 W m$^{-2}$) was found, and it was confirmed that the value was incorrect | The QC code was manually adjusted to 2 |
| | Ta | 2.8 m | 2009.11.25, 2009.11.26, 2015.06.07, 2017.05.14, 2020.05.25 | A few successive data points had values within the threshold range but their values were extremely low. There was a noticeable temperature difference between the 1.5 m and 2.8 m observations. | The QC codes were manually adjusted to 2 |
| | Rld | - | 2016.06.16-2019.10.25 | The seasonal variation of Rld in terms of daily mean, minimum, and maximum during this period were noticeably higher than in other periods | The QC codes were manually adjusted to 2 |
| NAMORS | ST | 0.1 m | 2008.10.16-2018.10.06 | The ST were much lower than those observed at the 0.2 m in all seasons, which was inconsistent with reality even though the data for this period were within the threshold range and shown significant seasonal variation. | The QC codes were manually adjusted to 1 |
| | ST | 0.1 m | 2019.09.12-2019.09.24 | A few successive data points had values within the threshold range but their values were extremely low. | The QC codes were manually adjusted to 1 |
| SETORS | Ta | 1.3 m | 2010.12.05-2010.12.08, 2016.10.31-2016.11.07, 2016.12.02-2016.12.29 | There was a noticeable temperature difference between the 1.3 m and 4.94 m. | The QC codes were manually adjusted to 2 |
| TGL | Rlu | - | 2018.05.29-2021.06.21 | Most of the Rlu values in this station are negative, the accuracy of the non-negative values cannot be guaranteed. | All the QC codes were manually adjusted to 1 |
| | Rld | - | 2018.05.29-2021.06.21 | Same as the Rlu. | All the QC codes were manually adjusted to 1 |

**Table 4. Mean error (ME), mean absolute error (MAE), root mean square error (RMSE), p-value from t test, and coefficient of determination calculated based on gap-filled artificially created missing data series and true values for gap lengths of 1, 2, and 3 hours, respectively.**

| | WS_1.5m | WD_1.5m | Ta_1.5m | RH_1.5m | Rsd | Rsu | ST_0.1m | ST_0.8m | H | LE |
|---|---|---|---|---|---|---|---|---|---|---|
| | | | | | Mean Error, ME | | | | | |
| Gap_1 | -0.011 | 1.439 | 0.001 | 0.077 | 0.501 | 0.180 | -0.002 | 0.0005 | 1.514 | 0.371 |
| Gap_2 | 0.011 | 1.139 | -0.012 | 0.123 | -1.331 | 0.588 | -0.003 | 0.0007 | 2.749 | -3.407 |
| Gap_3 | -0.009 | 2.331 | -0.042 | 0.143 | -1.358 | 2.158 | 0.006 | 0.0014 | 4.302 | -1.433 |
| | | | | | Mean Absolute Error, MAE | | | | | |
| Gap_1 | 0.816 | 57.342 | 0.596 | 3.013 | 46.515 | 15.917 | 0.069 | 0.0099 | 20.639 | 12.177 |
| Gap_2 | 0.929 | 63.531 | 0.894 | 4.023 | 67.617 | 20.714 | 0.206 | 0.0114 | 26.460 | 18.827 |
| Gap_3 | 1.159 | 71.068 | 2.350 | 8.423 | 148.017 | 39.256 | 0.754 | 0.0162 | 46.514 | 21.081 |
| | | | | | Root Mean Square Error, RMSE | | | | | |
| Gap_1 | 1.128 | 90.107 | 0.879 | 4.775 | 99.612 | 34.119 | 0.108 | 0.0267 | 33.564 | 27.244 |
| Gap_2 | 1.275 | 96.611 | 1.305 | 6.366 | 122.297 | 38.740 | 0.291 | 0.0281 | 40.256 | 212.681 |
| Gap_3 | 1.569 | 100.235 | 3.000 | 11.861 | 227.072 | 61.860 | 0.994 | 0.0388 | 64.444 | 70.102 |
| | | | | | P value from t test | | | | | |
| Gap_1 | 0.798 | 0.506 | 0.994 | 0.885 | 0.941 | 0.921 | 0.991 | 0.9961 | 0.571 | 0.837 |
| Gap_2 | 0.716 | 0.459 | 0.921 | 0.726 | 0.778 | 0.639 | 0.977 | 0.9926 | 0.158 | 0.453 |
| Gap_3 | 0.728 | 0.068 | 0.659 | 0.624 | 0.721 | 0.034 | 0.948 | 0.9822 | 0.005 | 0.569 |
| | | | | | Coefficient of determination, $R^2$ | | | | | |
| Gap_1 | 0.757 | 0.237 | 0.990 | 0.966 | 0.921 | 0.863 | 1.000 | 1.0 | 0.896 | 0.789 |
| Gap_2 | 0.692 | 0.167 | 0.977 | 0.940 | 0.881 | 0.821 | 0.999 | 1.0 | 0.861 | 0.259 |
| Gap_3 | 0.546 | 0.144 | 0.876 | 0.793 | 0.593 | 0.562 | 0.982 | 1.0 | 0.632 | 0.893 |

**Appendix A: Abbreviations used in text.**

**Met:** near-surface micrometeorological observations

**Soil:** soil hydrothermal profile observations

**Flux:** turbulent flux observations

**WS:** wind speed

**WD:** wind direction

**Ta:** air temperature

**RH:** relative humidity

**Pressure:** air pressure

**Rsd:** downward shortwave radiation

**Rsu:** upward shortwave radiation

**Rld:** downward shortwave radiation

**Rlu:** upward shortwave radiation

**ST:** soil temperature

**SM:** soil moisture

**H:** sensible heat flux

**LE:** latent heat flux

**QC:** quality control

**Appendix B: Data integrity and data quality report on a monthly scale.**

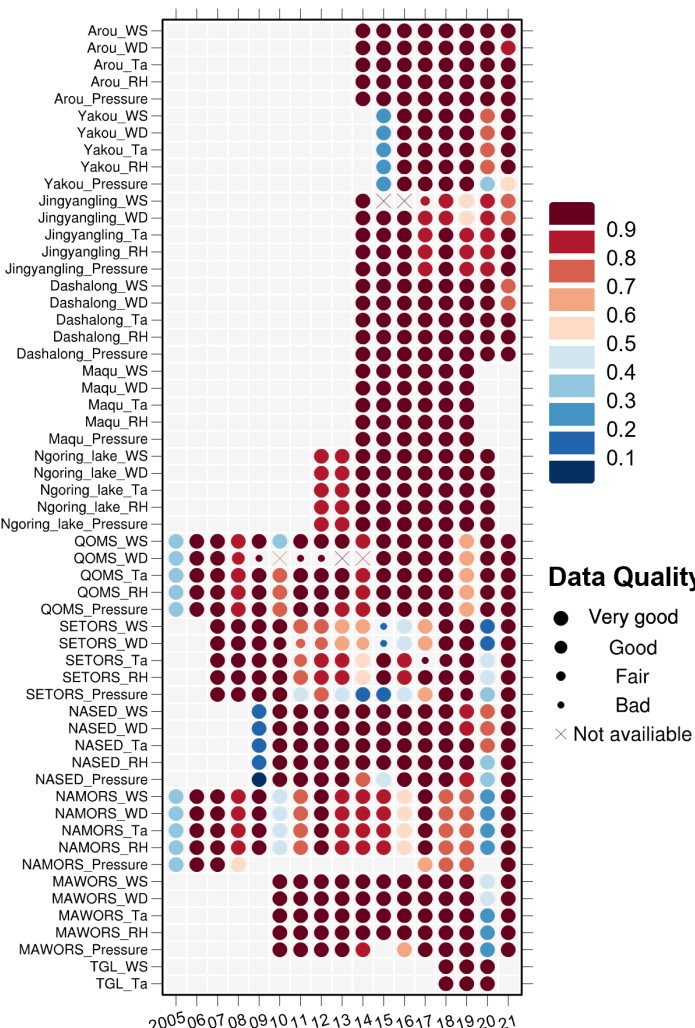

**Figure B1.** The data integrity and data quality report of the Met observations on a monthly scale. The color of the dots denotes the percentage of the available data on a monthly scale (blank indicates that all data for the month are not provided), while the size of the dots denotes the quality of these data. Bad: the percentage of correct data is less than 30%; Fair: the percentage of correct data is greater than 30% but less than 60%; Good: the percentage of correct data is greater than 60% but less than 80%; Very good: more than 80% of the data are correct; Not available indicates that all data for the month are not correct.

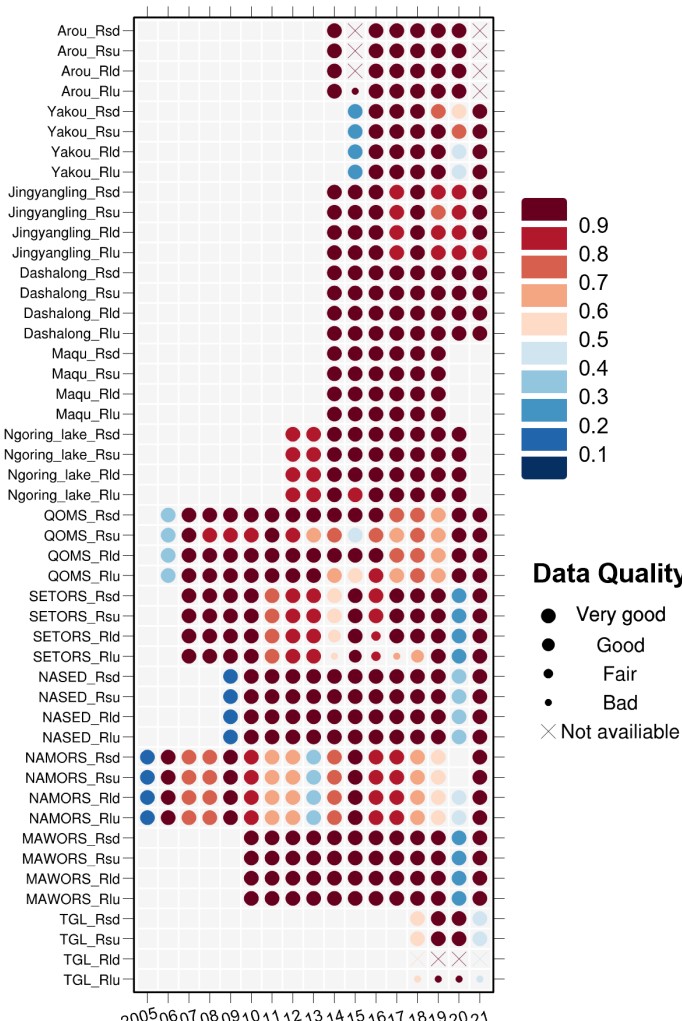

**Figure B2:** Same as the Fig. B1 but for the radiation measurements.

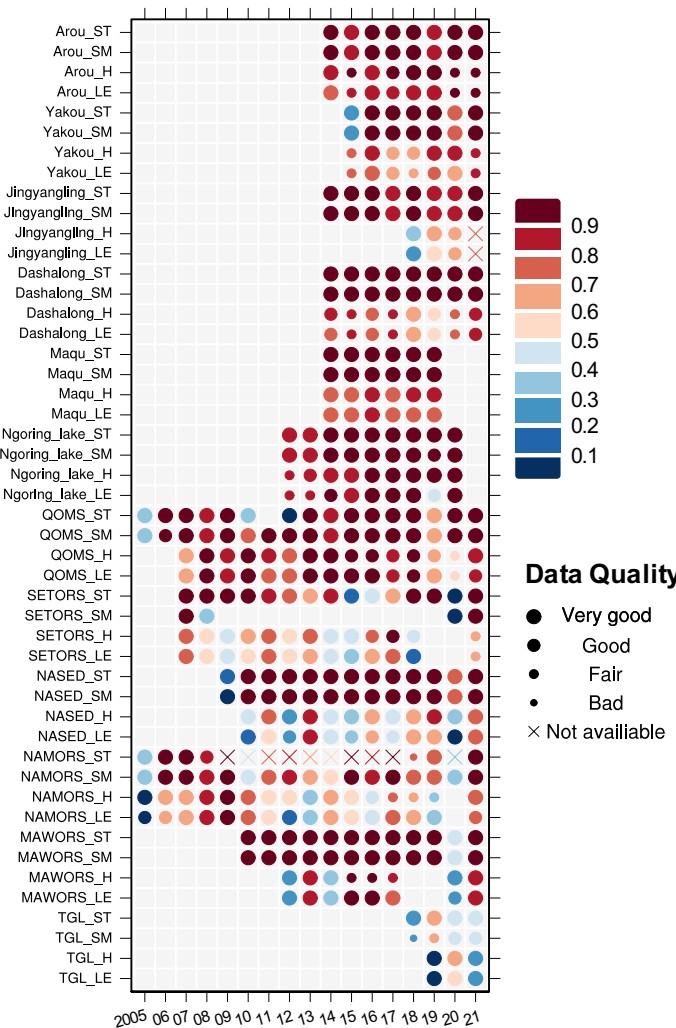

**Figure B3:** Same as the Fig. B1 but for the soil hydrothermal and turbulent flux measurements.