# Peer review of "Spatially Extensive Long-term Quality-assured Land-atmosphere Interactions Dataset over the Tibetan Plateau"

_Earth System Science Data, 2024_

## Referee Comment (RC1)

The manuscript presents a comprehensive dataset detailing land-atmosphere interactions over the Tibetan Plateau, derived from 12 field stations covering a range of landscapes. This dataset encompasses hourly measurements of surface energy balance components, soil hydrothermal properties, and near-surface micrometeorological conditions for up to 17 years (2005-2021). However, I have several major concerns that the authors should address.

1) Section 2 provides extensive detail on the observation infrastructure and data post-processing workflow, including data processing, quality control, gap filling, and archiving procedures. The authors should include more explicit information on the calibration of instruments across different stations and the rationale behind the selection of specific quality control algorithms. Comparisons with standard practices in the field could help in benchmarking the dataset's reliability.

2) The authors should provide a comprehensive and detailed explanation of the data collection methods and quality control procedures employed in their study. Instead of merely listing various methodologies, it is crucial to elaborate on how data was gathered, the criteria used for data selection, and the specific steps taken to ensure the integrity and accuracy of the data.

3) While the approach for handling missing data through linear temporal interpolation is mentioned in 2.3.3 Gap filling, a discussion on the impact of these interpolations on the dataset's overall quality and potential biases introduced should be mentioned. Including statistical metrics to quantify the robustness of the gap-filled data could enhance the dataset's credibility.

4) Section 3 on different datasets are well-detailed but the authors should add specific examples of data validation against external measurements or models, if available. This could include inter-comparison with satellite data, other observational networks, or model outputs to validate the spatial and temporal accuracy of the dataset.

---

## Author Comment (AC1)

**RC1: 'Comment on essd-2024-9'**

*The manuscript presents a comprehensive dataset detailing land-atmosphere interactions over the Tibetan Plateau, derived from 12 field stations covering a range of landscapes. This dataset encompasses hourly measurements of surface energy balance components, soil hydrothermal properties, and near-surface micrometeorological conditions for up to 17 years (2005-2021). However, I have several major concerns that the authors should address.*

**Response:** We are grateful to reviewer #1 for the effort reviewing our paper and the constructive feedback provided. Here below we tried our best to address all the concerns and suggestions raised by the reviewer #1. We hope that the modification made on the revised manuscript will cover the reviewer expectation. Changes highlighted in red have been made accordingly in the revised manuscript. The revised sentences are highlighted in blue in the following replies.

*1) Section 2 provides extensive detail on the observation infrastructure and data post-processing workflow, including data processing, quality control, gap filling, and archiving procedures. The authors should include more explicit information on the calibration of instruments across different stations and the rationale behind the selection of specific quality control algorithms. Comparisons with standard practices in the field could help in benchmarking the dataset's reliability.*

**Response:** Thank you very much for your careful review and insightful comment. The quality control procedures implemented in this study are standard practices following the guidelines described by Zahumenský (2004) as a proposal submitted to WMO. These quality control procedures are currently widely adopted to detect errors and ensure quality of meteorological data, despite varying degrees of modifications have been made to deal with site difference (e.g., different climate conditions, measured meteorological variables, and sensor hardware specifications) and specific issues. The design of the entire data quality control workflow in this study and some of the thresholds (for example, the plausible rate of change) used in the testing protocols are also followed the WMO's recommendation. Those standard quality controls of meteorological data, like checks for range limits and temporal consistency checks, are adjusted to the requirements of the instrumental and geographical settings of the research sites. This is due to the fact that generic methods frequently failed to deal with site-specific issues and unique problems that emerged from the field observations. The following texts were added to the original manuscript based on your suggestion.

■ Calibration instruments:

Line 214-228: Calibration of instruments is critical for ensuring accurate measurements. It is important to note, however, calibrating in a particularly harsh environment such as the TP is challenging. As a result, for meteorological and soil observations, both of which are relatively stable, calibrated reference instruments were used on a regular basis to perform field calibration across multiple stations, or the calibration was performed in a laboratory setting when instruments were returned for repair. In the case of turbulent observations, the measurement accuracy of the gas analyzer (i.e., LI-7500 and LI-7500DS) depends upon the cleanliness of the instrument lenses, it needs to be calibrated at regular intervals (once every six months at the five sites affiliated with

the ITPCAS) due to signal attenuation for $CO_2/H_2O$. The calibration consists of two major components: 1) determining the values of the calibration coefficients, and 2) adjusting zero and span to align the gas analyzer's actual response with the previously determined factory response. In addition, we conduct monthly inspections of the operational status of all observational equipment (Ma et al., 2023), as well as semi-annual on-site instrument maintenance for all stations, which includes instrument cleaning, checking the level of commissioned instruments, and checking instrument cables and connectors.

■ Selection of specific quality control algorithms:

Line 254-261: To provide the best level of accuracy feasible, an automatic processing scheme was specifically designed for each type of variable, following the guidelines described by Zahumensky (2004). Despite a wide array of methods has been proposed to obtain plausible micrometeorological data series, those methods share similar processing flow but varying degrees of modifications were made to deal with site-specific concerns and unique problems that emerged from the field observations. This is due to the fact that generic methods frequently failed to resolve these issues. This scheme is specifically adapted aimed at verifying the reliability of observations and detecting errors and suspicious values. The automatic data processing chain was built up as a series of sequential checks recommended by Zahumensky (2004), with emphasis on continuity and inter-consistency of meteorological fields to detect suspect observations.

*2) The authors should provide a comprehensive and detailed explanation of the data collection methods and quality control procedures employed in their study. Instead of merely listing various methodologies, it is crucial to elaborate on how data was gathered, the criteria used for data selection, and the specific steps taken to ensure the integrity and accuracy of the data.*

**Response:** We completely agree with the comment made by the Reviewer and really appreciate for pointing this issue out. We have supplemented the Section 2.2 with a detailed explanation of the data collection method used in our current field practice.

Line 225-228: To the maximum extent feasible, qualified personnel will take over and rectify any instrument malfunctions found during routine inspection (on-site or remote) to ensure the accuracy and integrity of the observations. Data logger (e.g., CR6, Campbell Scientific, USA) recordings are first temporarily stored on the memory card before being routinely transmitted to our Data Processing Center by wireless transmission or on-site collection for processing, analysis, and archiving.

We do understand your concerns regarding the quality control procedures. We tried to introduce the data quality control procedures in a better way by referring to relevant literatures (e.g., Fiebrich et al., 2010; Rollenbeck et al., 2016; Cerlini et al., 2020) on data quality control and assessment of meteorological observations, the following revisions have been made to the original manuscript to address the Reviewer's concerns.

- **Redesigning the Figure 2a.** To clearly demonstrate the workflow of the quality control procedures employed in this study, and for a detailed description of the criteria used to resolve the concerns raised from the Reviewer, we redesign the Figure 2a.
- **Adding the Figure 2b.** Figure 2b was added to clearly illustrate the key formulas, and

quantitative metrics used in each procedure.

- **Expanding description.** The following text was added in the revised manuscript to provide additional information on how the data flows through the procedures.

  Line 262-266: The program first reads the data file for each station to be processed, checks were performed sequentially from left to right, and only when all the prescribed check procedures for each variable completed before moving on to the next one. The quality control procedures are arranged in a deliberate sequence, and ignore values flagged as errors by preceding checks in the sequence because the checks each have specific data requirements (e.g., running average and corresponding standard deviation should be calculated based on correct data).

**a). Post-processing workflow**

[Figure]

**b). Quality control**

Aside from interpolation data with short-term gaps, we did not take any specific steps to adjust the data series during the post-processing phase. Only short gaps were filled because the performance of the reconstruction method is strictly dependent on the length of the data gap, long-term gaps may greatly affect the reliable and accuracy of the observations. The main purpose of our data quality control is to identify and locate problems in the data series and flag them so that data users can base their research on reliable observations. To ensure the integrity, continuity, and reliable of the observations, we primarily implement targeted actions. For instance, we pay close attention to the operational status of the equipment throughout field observations and promptly address any issues that arise, such as instrument malfunctions or abnormal data. The following content has been added in the revised manuscript to describe the efforts we have made to ensure the integrity and accuracy of the data.

Line 222-226: In addition, we conduct monthly inspections of the operational status of all

observational equipment (Ma et al., 2023b), as well as semi-annual on-site instrument maintenance for all stations, which includes instrument cleaning, checking the level of commissioned instruments, and checking instrument cables and connectors. To the maximum extent feasible, qualified personnel will take over and rectify any instrument malfunctions found during routine inspection (on-site or remote) to ensure the accuracy and integrity of the observations.

We think these revisions now provide a more comprehensive and detailed explanation of the data collection methods and quality control procedures, and we hope that the revision is acceptable and the Reviewer feel satisfied with this revision.

[1] Cerlini P B, Silvestri L, Saraceni M. Quality control and gap-filling methods applied to hourly temperature observations over central Italy[J]. Meteorological Applications, 2020, 27(3): e1913.

[2] Fiebrich C A, Morgan C R, McCombs A G, et al. Quality assurance procedures for mesoscale meteorological data[J]. Journal of Atmospheric and Oceanic Technology, 2010, 27(10): 1565-1582.

[3] Rollenbeck R, Trachte K, Bendix J. A new class of quality controls for micrometeorological data in complex tropical environments[J]. Journal of Atmospheric and Oceanic Technology, 2016, 33(1): 169-183.

*3) While the approach for handling missing data through linear temporal interpolation is mentioned in 2.3.3 Gap filling, a discussion on the impact of these interpolations on the dataset's overall quality and potential biases introduced should be mentioned. Including statistical metrics to quantify the robustness of the gap-filled data could enhance the dataset's credibility.*

**Response:** You have raised an important question. The accuracy of the linear temporal interpolation based gap filling technique was assessed based on filling artificially generated data gaps with different lengths. The performance of the gap filling method and the robustness of the gap-filled data for gap lengths of 1, 2, and 3 hours was evaluated. We randomly select 5,000 records of wind speed, wind direction, air temperature, relatively humidity, downward shortwave radiation, upward shortwave radiation, soil temperatures (at depths of 0.1 m and 0.8 m), sensible heat flux and latent heat flux, respectively. These variables were selected for assessment because they exhibit varying degrees of variability in the observed values over relatively short intervals. For example, wind speed and wind direction vary significantly over 1-3 hours, but soil temperature exhibits litter variation in the same time frame. This gives a good illustration of the impact of the interpolation scheme on the variables with varying degrees of variability. The mean error, mean absolute error, root mean square error, p value from the t test, and r square were calculated and provided in the new added Table 4. The following content has been added in the revised manuscript based on your suggestion.

Line 322-335: Series of random gaps (5,000 records for each variable) with different lengths were artificially created to quantify the overall performance of the gap filling method used and the robustness of the gap-filled data produced. The performance in filling gaps in wind speed, wind direction, air temperature, relatively humidity, downward/upward shortwave radiation, soil temperatures (at depths of 0.1 and 0.8 m), sensible heat flux, and latent heat flux for gap lengths

of 1, 2, 3 hours were evaluated. These variables were selected for assessment because they exhibit varying degrees of variability in the observed values over relatively short intervals. For example, wind speed and wind direction vary significantly over 1-3 hours, whereas soil temperature changes less during that time. Table 4 shows the mean error, mean absolute error, root mean square error, p value from the t test, and r square. Results suggest that the gap length is one of the key factors influences the performance. This is demonstrated by the fact that the longer the gap length, the greater the error (ME, MAE, and RMSE) and the lower the coefficient of determination of the regression between the real values and the gap filled values, as well as the relatively larger errors of the variables with a higher degree of variability in a short period of time (wind direction, for example, is the most unreliable to interpolate). The interpolated upward shortwave radiation series with three hours gaps differs significantly ($p<0.05$) from the true values, for other variables evaluated, the difference is not significant. These findings suggest that the gap filling method used in this study can reasonably reconstruct the gaps within one to three hours.

Table 4. Mean error (ME), mean absolute error (MAE), root mean square error (RMSE), p-value from t test, and coefficient of determination calculated based on gap-filled artificially created missing data series and true values for gap lengths of 1, 2, and 3 hours, respectively.

| | WS_1.5m | WD_1.5m | Ta_1.5m | RH_1.5m | Rsd | Rsu | ST_0.1m | ST_0.8m | H | LE |
|---|---|---|---|---|---|---|---|---|---|---|
| Mean Error, ME | | | | | | | | | | |
| Gap_1 | -0.011 | 1.439 | 0.001 | 0.077 | 0.501 | 0.180 | -0.002 | 0.0005 | 1.514 | 0.371 |
| Gap_2 | 0.011 | 1.139 | -0.012 | 0.123 | -1.331 | 0.588 | -0.003 | 0.0007 | 2.749 | -3.407 |
| Gap_3 | -0.009 | 2.331 | -0.042 | 0.143 | -1.358 | 2.158 | 0.006 | 0.0014 | 4.302 | -1.433 |
| Mean Absolute Error, MAE | | | | | | | | | | |
| Gap_1 | 0.816 | 57.342 | 0.596 | 3.013 | 46.515 | 15.917 | 0.069 | 0.0099 | 20.639 | 12.177 |
| Gap_2 | 0.929 | 63.531 | 0.894 | 4.023 | 67.617 | 20.714 | 0.206 | 0.0114 | 26.460 | 18.827 |
| Gap_3 | 1.159 | 71.068 | 2.350 | 8.423 | 148.017 | 39.256 | 0.754 | 0.0162 | 46.514 | 21.081 |
| Root Mean Square Error, RMSE | | | | | | | | | | |
| Gap_1 | 1.128 | 90.107 | 0.879 | 4.775 | 99.612 | 34.119 | 0.108 | 0.0267 | 33.564 | 27.244 |
| Gap_2 | 1.275 | 96.611 | 1.305 | 6.366 | 122.297 | 38.740 | 0.291 | 0.0281 | 40.256 | 212.681 |
| Gap_3 | 1.569 | 100.235 | 3.000 | 11.861 | 227.072 | 61.860 | 0.994 | 0.0388 | 64.444 | 70.102 |
| P value from t test | | | | | | | | | | |
| Gap_1 | 0.798 | 0.506 | 0.994 | 0.885 | 0.941 | 0.921 | 0.991 | 0.9961 | 0.571 | 0.837 |
| Gap_2 | 0.716 | 0.459 | 0.921 | 0.726 | 0.778 | 0.639 | 0.977 | 0.9926 | 0.158 | 0.453 |
| Gap_3 | 0.728 | 0.068 | 0.659 | 0.624 | 0.721 | 0.034 | 0.948 | 0.9822 | 0.005 | 0.569 |
| Coefficient of determination, $R^2$ | | | | | | | | | | |
| Gap_1 | 0.757 | 0.237 | 0.990 | 0.966 | 0.921 | 0.863 | 1.000 | 1.0 | 0.896 | 0.789 |
| Gap_2 | 0.692 | 0.167 | 0.977 | 0.940 | 0.881 | 0.821 | 0.999 | 1.0 | 0.861 | 0.259 |
| Gap_3 | 0.546 | 0.144 | 0.876 | 0.793 | 0.593 | 0.562 | 0.982 | 1.0 | 0.632 | 0.893 |

*4) Section 3 on different datasets are well-detailed but the authors should add specific examples of data validation against external measurements or models, if available. This could include inter-comparison with satellite data, other observational networks, or model outputs to validate the spatial and temporal accuracy of the dataset.*

**Response:** Thank you for this valuable suggestion. Unfortunately, in-situ field observations are

extremely rare in such an extreme environment as the Tibetan Plateau. The closest automatic weather station operated by China Meteorological Administration (CMA) to those in our dataset are tens to hundreds of kilometers away. Additionally, the ASWs typically record only conventional meteorological elements, soil hydrothermal and turbulent fluxes are not available. Due to the difference in topography and subsurface features, observations between field stations and CMA operational AWSs may differ dramatically. Our stations included in the dataset represent currently the only observational network in the TP region that is able to conduct comprehensive observational measurements of land-atmosphere interactions. It is not suggested to validate the in-situ observations against model results, reanalysis products, or remote sensing products because of their poor accuracy in this area, which can be attributed to several factors like resolution. Besides, a great deal of work has been done on the assessment of model results and remote sensing products based on the observations provided in our dataset (e.g., Minola et al., 2024; Yao et al., 2023; Tong et al., 2023). Taking these factors into account, we therefore did not use external measurements or models to validate the in-situ observations. But your suggestion has motivated us to investigate the possibility of using multi-source data from satellite products, model outputs, and reanalysis products to filling the gap data in the field observations in future work. This is a better approach than the linear interpolation method we used in this study, and it can be applied to a wider range of scenarios. We hope you could understand our concern. Thank you very much.

[1]	Minola L, Zhang G, Ou T, et al. Climatology of near-surface wind speed from observational, reanalysis and high-resolution regional climate model data over the Tibetan Plateau[J]. Climate Dynamics, 2024, 62(2): 933-953.

[2]	Yao T, Lu H, Yu Q, et al. Uncertainties of three high-resolution actual evapotranspiration products across China: Comparisons and applications[J]. Atmospheric Research, 2023, 286: 106682.

[3]	Tong L, He T, Ma Y, et al. Evaluation and intercomparison of multiple satellite-derived and reanalysis downward shortwave radiation products in China[J]. International Journal of Digital Earth, 2023, 16(1): 1853-1884.

---

## Author Comment (AC2)

**Response to the RC2**

**RC2: 'Comment on essd-2024-9'**

*The paper by Ma et al. focuses on generating in situ records relating to land-atmosphere interactions through an integrated observations network across the Tibetan Plateau. This work is immensely important for understanding the behavior of atmospheric boundary layer across various landscapes over the Tibetan Plateau, where site observations are notably scarce. Moreover, those measurements can be used for calibrating and assessing land surface models and remote sensing observations. The following comments warrant attention.*

**Response:** Special thanks to you for these insightful comments. In the revised version of the manuscript, we have addressed essentially all the points raised. We hope that the modification made on the revised manuscript will cover the reviewer expectation. We appreciate the positive comments highlighting the contributions of our work. Our ongoing goals are to guarantee the accessibility and accuracy of these field observational data and to offer solid data support for the study of climate change and its environmental effects of the Tibetan Plateau. The revised contents are highlighted in blue in the following responses, corresponding changes are marked in red in the revised manuscript.

*1. Abstract needs to be concise. The first two sentences had provided background information, please delete the sentence 'The TP is recognized … with diverse landscape'. Remove the content 'Scientific data sharing is critical for the TP … they bring about' into main text. Include more information about which kind of variable you are going to provide and temporal extent.*

**Response:** It is really true as Reviewer suggested that the language in the Abstract is not concise enough. We have made a thorough modification to the abstract to improve it. The revised abstract is listed as follows. We gratefully appreciate for your advice.

Abstract: The climate on the Tibetan Plateau (TP) has experienced substantial changes in recent decades as a result of its susceptibility to global climate change. The changes observed across the TP are closely associated with regional land-atmosphere interactions. Current models and satellites struggle to accurately depict the interactions, critical field observations on land-atmosphere interactions here therefore provide necessitate independent validation data and fine-scale process insights for constraining reanalysis products, remote sensing retrievals, and land surface model parameterizations. Scientific data sharing is crucial for the TP since in-situ observations are rarely available in this harsh condition. However, field observations are currently dispersed among individuals or groups and have not yet been integrated for comprehensive analysis. This has prevented a better understanding of the interactions, the unprecedented changes they generate, and the substantial ecological and environmental consequences they bring about. In this study, we collaborated with different agencies and organizations to present a comprehensive dataset for hourly measurements of surface energy balance components, soil hydrothermal properties, and near-surface micrometeorological conditions spanning up to 17 years (2005-2021). This dataset, derived from 12 field stations covering a variety of typical TP landscapes, provides the most extensive in-situ observation data available for studying land-atmosphere interactions on the TP to date in terms of both spatial coverage and duration. Three categories of observations are provided

in this dataset: meteorological gradient data (Met), soil hydrothermal data (Soil), and turbulent flux (Flux). To assure data quality, a set of rigorous data processing and quality control procedures are implemented for all observation elements (e.g., wind speed and direction at different height) in this dataset. The operational workflow and procedures are individually tailored to the varied types of elements at each station, including automated error screening, manual inspection, diagnostic checking, adjustments, and quality flagging. The hourly raw data series, the quality-assured data, and supplementary information including data integrity and the percentage of correct data on a monthly scale are provided via the National Tibetan Plateau Data Center (https://doi.org/10.11888/Atmos.tpdc.300977, Ma et al., 2023). The present dataset provides the benchmark constraints needed to evaluate and refine the land surface models, reanalysis products, and remote sensing retrievals. It can also be used to characterize fine-scale land-atmosphere interaction processes of the TP, as well as underlying influence mechanisms.

*2. Section 2.1 and 2.2: Please provide a table in which each row represents one site and each column include one unique information. Then please provide the site name, location, climate, landscape type, installation of infrastructure, and measuring variables. If it is too large. It would be OK to provide two tables. One for basic information and another for introducing infrastructure installation, managing period, and measuring variables. Please provide as much details as you can for publishing a data paper.*

**Response:** We agree with the reviewer in this regard, and we also believe that a data descriptor paper should include the table to facilitate the data users grasp the dataset as soon as possible. In fact, we have taken this into account when preparing the first draft of this manuscript, the Table 1 (provided after the references) presents not only the basic information about each station (e.g., latitude, longitude, elevation, landscape type), but also the observation infrastructure (sensor model, manufacture, height, units, and observing period of each variable). The design of this table was informed by several previous papers that were also published in the ESSD. We apologize for placing this table after the references because it was too long, so you may not have noticed it.

*3. Section '2.3 Data post-processing workflow' needs further improvements.*
*(1) Figure 2: The information provide in this figure is a little bit general. It should be a summary of section 2.3.1 to 2.3.4. (i) We need to know the specific variables you are working on. (ii) Are you using those data processing approach for all variables? (iii) In the four modules, are you consistently applied these processing approaches to each variable and each site? I highly recommend that the author refer to previously published ESSD or other high-quality data papers and redesign the flowchart accordingly. I have provided the following paper for reference. Please note that there is no need to cite them.*
*Gebrechorkos, S. H., Peng, J., Dyer, E., Miralles, D. G., Vicente-Serrano, S. M., Funk, C., . . . Dadson, S. J. (2023). Global high-resolution drought indices for 1981–2022. Earth Syst. Sci. Data, 15(12), 5449-5466.*
*Pastorello, G., Trotta, C., Canfora, E. et al. The FLUXNET2015 dataset and the ONEFlux processing pipeline for eddy covariance data. Sci Data 7, 225 (2020).*
*Beck, H. E., E. F. Wood, M. Pan, C. K. Fisher, D. G. Miralles, A. I. J. M. van Dijk, T. R. McVicar, and R.*

*F. Adler, 2019: MSWEP V2 Global 3-Hourly 0.1° Precipitation: Methodology and Quantitative Assessment. Bull. Amer. Meteor. Soc., 100, 473–500.*

**Response:** Thank you very much for the above suggestions. Following your suggestion, we have redesigned the Figure 2a (flowchart of data post-processing workflow), the new figure is listed as follows. Besides, the Figure 2b was added to clearly show relevant information used in the data quality control procedures.

[Figure]

*(2) Section 2.3.1 to 2.3.4 require much more details: (i) Please list the relevant methods (equation, models, quantification metrics, etc) you used where are applicable. (ii) Definition of missing data should be quantified for each variable and each site. (iii) Provide a detailed description of the data header file format. Overall, this part is very important and much more details should be provided.*

**Response:** (i) We have carefully addressed all the reviewer's concern about the description of the data quality control. The Figure 2b was added to list all the relevant information (equation, metrics, and threshold values) used in data quality control. Figure 2b is a further extension of the Figure 2a, and we think the revised Figure 2 can give the reader a systematic and in-depth understanding of the entire data post-processing process applied in this work.

(ii) We apologize for any misunderstanding you may have experienced. Instead of using NAN or -9999 to indicate missing data as they were in the raw data, we intended to use 9999.9. To prevent misunderstanding, we thus replace "Definition of missing data" to "Missing value assignment" in the Data control step in Figure 2. Thanks very much for your understanding.

(iii) Based on your suggestion, we have modified the description of the data header. In the revised manuscript, we described the naming format of data header in Section 2.3.4 Data archiving and Section 5 Data availability. The revised description is listed as follows:

- Section 2.3.4 (Line 349-351): During the archiving step, the header descriptions of the output files were first standardized to include information about the variable name, height/depth, and units. This information was expressed in the following format: variable_height/depth(units). The variable names are expressed as abbreviations which are listed in Appendix A.
- Section 5(Line 553-555): The data header of each CSV-formatted data file contains comprehensive information on the variable units and heights/depths, with naming format: variable_height/depth (units). The variable names are expressed as abbreviations. Appendix A provides a full list of the abbreviations for each variable.

*4. Section 3 Data description: Much more details should be provided. Provide a table and listed all those variables this data set will provide. Indicate availability of each variable at a specific site. Provide unit for each variable and start date and end date (if applicable). The primary principle is assisting the data user quickly know how those valuable measurements fit their research.*

**Response:** Thanks very much for pointing this out. We do understand your concern regarding the data description. Table 1 (apologize once again) summarized all the variables that included in the dataset, providing the necessary information of each variable to show the units of the variables, the heights and periods these variables observed, the models and manufactures these sensors used. Furthermore, the available period of each variable can be clearly observed from the Figure B1-B3 which were shown as Appendix. We think this information can help the data users identify which variables they need.

*5. Section 4: it would be great if the authors can provide some application cases.*

**Response:** This suggestion is highly appreciated. We searched for recent studies that directly using in-situ observations from the stations included in the current dataset. As examples of the application cases of the in-situ observations, we have selected a few representative examples of studies in the fields of fine-scale interactions analysis, model representation assessment, model development, remote sensing algorithm refinement, and key land surface parameter estimation. The updated content is listed as follows with newly added sentences highlighted in blue.

- **Fine-scale interactions analysis**
  Line 525-529: More specifically, targeted field campaigns across the vast grasslands and permafrost zones are indispensable for capturing the fine-scale interactions between the changing land surface and the overlying atmosphere. This is exemplified by the studies conducted by Li et al., (2015) and Wang et al., (2019), who investigated the lake-atmosphere interactions using in-situ observations from the Ngoring Lake and NAMORS station, respectively.
- **Model representation assessment**
  Line 529-532: It is possible to systematically verify model representations of hydrological and

thermal processes, as well as their interconnections, at various sites using this enhanced monitoring network. The work done by Liu et al. (2018), which evaluated the effectiveness of the WRF model in snowfall simulation using in-situ measurements, serves as an example of this.

- Model development

  Line 532-535: It will serve as pillars for improving model physics concerning cryospheric, hydrologic, and atmospheric processes in the intricate TP terrain. An example of this is the study done in 2013 by Chen et al., whereby a DEM-based radiation model was developed for an accurate estimation of instantaneous clear sky solar radiation using measurements from the QOMS station.

- Remote sensing algorithm refinement

  Line 535-538: Meanwhile, comprehensive field measurements are crucially needed to validate, calibrate, and refine remote sensing retrieval algorithms over the topographically complex terrain. For instance, Yuan et al., (2021) used in-situ measurements from this dataset to present an optimized canopy transpiration model and an improved technique for calculating soil evaporation with soil moisture and texture.

- Key land surface parameter estimation

  Line 538-541: Systemic biases in key land surface parameters in the reanalysis products can be decreased by incorporating synthesized ground-based datasets and revised satellite products through sophisticated data assimilation techniques. For instance, Qi et al., (2023) increased the accuracy of land surface temperature retrieval over the TP based on the in-situ data.

---

## Author Comment (AC3)

**Response to the RC3**

**RC3: 'Comment on essd-2024-9'**

*This manuscript provides an overview of in-situ observations of land-atmosphere interactions at 12 unique sites across the Tibetan Plateau (TP). The authors first identify and describe the standard flux tower (e.g., EC, meteorology, soil) measurements collected at each site (types of instruments and heights) and then outline the quality control and quality assurance processes that are completed, before examining the seasonal and diurnal trends between each site. The work is important and novel. I have a few general comments:*

**Response:** We thank the reviewer for the attentive reading of our manuscript and the positive feedback. According to your nice suggestions, we have made extensive corrections to our previous manuscript, the detailed corrections are listed below. The revised contents are highlighted in blue in the following responses, corresponding changes are marked in red in the revised manuscript.

*1.) Introduction - The introduction follows a logical framework: importance of TP with regards to Earth system interactions, how the the TP is warming faster than other areas (and the implications), importance of models and datasets for decision making, challenges with model data inputs due to scarcity of in-situ observations, past efforts, and then potential issues (QA/QC of data) with open access datasets, but in it's current state it is a bit long (mainly the first, third, and fifth paragraphs). I would recommend trimming the introduction if possible.*

**Response:** Thank you very much for pointing out the problem in our introduction section. We thoroughly reviewed the content and realized that it is indeed a bit too long. Based on your recommendation, the following less important texts have been deleted in the revised manuscript to make the paragraph as brief as possible, while the overall logical framework remains unchanged.

Paragraph 1: These changes are significant and highly visible, while others, like shrinking permafrost areal extent (Ran et al., 2018), melting ground-ice (Chen et al., 2020), extensive thermokarst development (Luo et al., 2022), and shifting precipitation patterns (Yao et al., 2022), are typically more gradual and less obvious but still detectable (Thornton et al., 2021). Worsened desertification (e.g., Xue et al., 2009), enhanced terrestrial evapotranspiration (e.g., Ma and Zhang, 2022), rapid lake expansion (e.g., Zhang et al., 2021), and altered river discharges (e.g., Cao et al., 2006) are typically associated with the accelerated climate change.

Paragraph 3: For instance, declining glaciers and seasonal snow cover decrease surface albedo, raise solar radiation absorption, and promote further warming (Ghatak et al., 2014). This coupling between the land surface and atmosphere acts as feedback, exacerbating regional warming and hydroclimatic changes (Zhou et al., 2019).

Paragraph 5: These efforts are in accordance with international requests for open TP data to maximize the potential value of scientific data in broad applications and to advance scientific understanding of the interactions and feedback between the land and atmosphere… where the harsh environment itself poses fundamental threats to observation quality.

*2.) Observation Network and Data Processing - Similar to some of the other referee comments, I would like to see more specific details outlining the typical on-site calibrations and maintenance of instruments at each site and better address how you compare measurements at varying heights between sites (e.g., from Table 1 - EC heights ranging from 3 to 4.5 m, and met observations from 1.5 m, 2.75m, or 5 and 10 m).*

**Response:** This is an excellent suggestion. We highly value the concerns of the reviewers regarding the maintenance and calibration of the instruments. According to all the comments and suggestions, we tried our best to supplement this section, which outlines the on-site calibrations and maintenance of instruments at each site. The following content has been added in the revised manuscript. We hope that the modification made on the revised manuscript will cover the reviewer expectation.

Line 214-228: Calibration of instruments is critical for ensuring accurate measurements. It is important to note, however, calibrating in a particularly harsh environment such as the TP is challenging. As a result, for meteorological and soil observations, both of which are relatively stable, calibrated reference instruments were used on a regular basis to perform field calibration across multiple stations, or the calibration was performed in a laboratory setting when instruments were returned for repair. In the case of turbulent observations, the measurement accuracy of the gas analyzer (i.e., LI-7500 and LI-7500DS) depends upon the cleanliness of the instrument lenses, it needs to be calibrated at regular intervals (once every six months at the five sites affiliated with the ITPCAS) due to signal attenuation for $CO_2/H_2O$. The calibration consists of two major components: 1) determining the values of the calibration coefficients, and 2) adjusting zero and span to align the gas analyzer's actual response with the previously determined factory response. In addition, we conduct monthly inspections of the operational status of all observational equipment (Ma et al., 2023), as well as semi-annual on-site instrument maintenance for all stations, which includes instrument cleaning, checking the level of commissioned instruments, and checking instrument cables and connectors. To the maximum extent feasible, qualified personnel will take over and rectify any instrument malfunctions found during routine inspection (on-site or remote) to ensure the accuracy and integrity of the observations. Data logger (e.g., CR6, Campbell Scientific, USA) recordings are first temporarily stored on the memory card before being routinely transmitted to our Data Processing Center by wireless transmission or on-site collection for processing, analysis, and archiving.

As for your concern about the varying heights between sites, we tried to use observations at the same height/depth as much as possible in our current comparative analysis. For example, the 0.1 m depth soil hydrothermal variations were compared except for the NASED station (0.2 m depth observations were used because observations at 0.1 m depth were not recorded prior to 2020, line 549-550). Since the primary purpose of the comparison was to show the micrometeorological characteristics at the near-surface layer, therefore, height adjustment was not implemented. We compare the variations at the lowest level of each site directly with varying heights between stations. it is imperative to acknowledge that the differences in observing height across the stations do affect the comparison. Surface roughness length and the vertical lapse rate of air temperature are required when adjusting observing heights, this may introduce additional uncertainty. Furthermore, sensible heat flux and latent heat flux are highly depended on the source area, which increases with observing height. This would require in-depth analysis of the flux contribution

source region distribution, which is somewhat outside the scope of this paper. The following modifications have been made to the original manuscript to clarify the site comparison.

Line 359-360: Note that the lowest layer was chosen for stations with gradient meteorological parameters observed, although observing height vary among stations, height adjustment is not implemented in this study.

Line 450: Fig. 7 compares the multi-year mean diurnal and seasonal variations of the shallow-layer (0.1 m depth) soil temperature and soil moisture to better illustrate the hydrothermal differences due to the spatial variability in soil physical and chemical properties (e.g., soil type, porosity, organic matter content), vegetation characteristics, and meteorological conditions between stations.

Line 468-470: Figure 7. Seasonal variations of the diurnal (the first and the third column) and daily mean (the second and the fourth column) shallow-layer (0.1 m depth for stations except the NASED where 0.2 m was used) soil temperature (a-d), and soil water content (e-h) at the 12 stations.

*3.) Eddy Covariance Data - Were there any differences found between the LI-7500s and the EC150 at Maqu? Was this examined? You might cite a supporting paper to address this if applicable. Also, skipping a bit ahead, but in Figure B3, all of the sensible heat (H) data are marked as 'bad' data quality. Why is this? Why are these data still considered/highlighted in the manuscript if they are so bad (Figure 8) ? Similarly, how can there be very good LE data but bad H data if they are both being derived from the H2O flux in the EC setup? Please address.*

**Response:** You have raised an important question. Unfortunately, we do not have these two different types of gas analyzers installed at Maqu station to test the comparability of the turbulent fluxes. After searching the literature, we discovered that Frank and Massman conducted a careful comparative analysis of seven distinct kinds of fast-response hygrometers including open-path (e.g., LI-7500, EC150) and closed-path (e.g., LI-7000, LI-7200, and EC155) analyzers, results show that "there was minimal evidence to support that water vapor flux measurements are meaningfully different among common hygrometers in use today, as well as historically important sensors". Another study conducted by Polonik et al., (2019) reports that "all sensors, regardless of type, can be used to measure fluxes if appropriate corrections are applied and quality control measures are taken".

[1] Polonik P, Chan W S, Billesbach D P, et al. Comparison of gas analyzers for eddy covariance: Effects of analyzer type and spectral corrections on fluxes[J]. Agricultural and Forest Meteorology, 2019, 272: 128-142.
[2] Frank J M, Massman W J. A study of the role of seven historically significant fast-response hygrometers and sensor calibration on eddy covariance H2O fluxes and surface energy balance closure[J]. Agricultural and Forest Meteorology, 2023, 334: 109437.

We sincerely thank you for your careful checks on the Figure 3B. We apologize for the mistake of loading the wrong data when drawing Figure 3B. We have redrawn the figure and double-checked the data for other figures in the Appendix to ensure a problem-free manuscript. Both H and LE are good in the dataset. Thank you again for pointing this issue out.

*4.) Data Descriptions - I have some general questions/comments about Section 3. Could the higher nighttime wind speeds at Yakou be attributed to the higher measurement height (10 m at that site vs 1 m at other sites)? What benefit do the pressure data provide given the different site altitudes? Can you comment on the diurnal offset in H and LE at Jingyangling (Figure 8)? All others sites in Figure 8 follow a similar trend, except for Jingyangling, does this mean H and LE are peaking at night? Lastly, since this is a data paper, it might be better to forgo the results and site comparisons outlined in much of Section 3, and instead provide a brief comparison of how these in-situ data stack up against aforementioned model or remote sensing data within the TP.*

**Response:** We sincerely thank the reviewer for careful reading. The response to each of the above questions are listed as follows:

■ The higher nighttime wind speed at Yakou station is mainly due to the unique topography of the station (located on the highland). The wind speeds at the Arou, Jingyangling and Dashalong stations are measured at a height of 10 m.

■ It can be seen from the comparison of pressure that the air pressure is highly dependent on the site altitude, and the variations in barometric pressure between stations with small difference in altitude were essentially the same. Therefore, it may be further considered to use the barometric formula and combined with air pressure observations measured at nearby station to perform data quality control and gap filling for the time series of barometric pressure.

■ We regret to admit that, although some of the data recorded in 2021 did pass the tests, we believe that the quality of the data for this period cannot be fully guaranteed due to the excessively large values of turbulent fluxes at night (a plausible reason for this could be related to the timestamp). This is based on a careful analysis and evaluation of the turbulent flux observations. This resulted in the abnormally high values of the nighttime turbulent fluxes in the diurnal variation at the Jingyangling station. We have manually adjusted the QC code to 2 to guarantee the accuracy of the observations and to prevent this portion of data from being misused in subsequent analysis and research. Once the problematic observations discarded, the variation of turbulent fluxes are consistent with other stations.

■ We express our gratitude for your insightful suggestion and comment to the Section 3. After discussion, we think that the site comparison of the observation variables is crucial, primarily for the following reasons. This preliminary comparative analysis provides initial insights into topographic influences, seasonal cycles, interannual variability, and spatial heterogeneity that can be explored in greater depth through focused studies using this multi-site dataset. It can also be a very good way to demonstrate data quality and potential scientific value by locating differences among stations and special variations (e.g., the positive nighttime turbulent fluxes observed at the Jingyangling station). Furthermore, a great deal of work has been done on the assessment of model results and remote sensing products based on the observations provided in our dataset (e.g., Minola et al., 2024; Yao et al. 2023; Tong et al., 2023), while comparative analysis is rare, this is one important reason we did not compare the field observations with the model results and remote sensing products. Meanwhile, following the general practice widely used in some previous articles published in this journal describing the field observation dataset, the site comparisons outlined in Section 3 is retained in the revised

manuscript. We appreciate your understanding. Thank you.

[1] Minola L, Zhang G, Ou T, et al. Climatology of near-surface wind speed from observational, reanalysis and high-resolution regional climate model data over the Tibetan Plateau[J]. Climate Dynamics, 2024, 62(2): 933-953.

[2] Yao T, Lu H, Yu Q, et al. Uncertainties of three high-resolution actual evapotranspiration products across China: Comparisons and applications[J]. Atmospheric Research, 2023, 286: 106682.

[3] Tong L, He T, Ma Y, et al. Evaluation and intercomparison of multiple satellite-derived and reanalysis downward shortwave radiation products in China[J]. International Journal of Digital Earth, 2023, 16(1): 1853-1884.

---

## Referee Report (RR1)

Thank you for the revisions made to the manuscript based on previous feedback. The manuscript has seen considerable improvement, but there are still areas that require further attention to ensure clarity and comprehensive presentation.

1) L58-59: Consider adding specific impacts or key findings that underline the importance of this new dataset, not just use this simplified sentence.

2) L68-69: Specific the type of "evident changes" observed in the TP (e.g., temperature changes, precipitation patterns) to provide more context and clarity for the reader.

3) L113-114: It would be beneficial to add more discussions about existing datasets and their limitations to contextualize the necessity of the new dataset. This comparison will help in understanding the incremental or significant advancements made.

4) The methodology section would benefit from more detailed descriptions of the statistical tests to analyze the data. Clarification on the choice of these methods and their appropriateness for your dataset would strengthen this section.

5) In Section 4 (L520-545), adding comparative analysis with prior data from other studies or contrasting regions will help in emphasizing the unique contributions of this new dataset.

6) The conclusion reiterates much of the information from the abstract and introduction and needs to be rewritten. To add value, consider highlighting the most advantages of this new dataset, the important findings from your dataset and specific implications for future research or policy changes prompted by this new dataset.

---

## Referee Report (RR2)

Most of the previous concerns have been adequately addressed. The following issues require attention:

1. Please ensure consistent use of scientific notation throughout the manuscript. For example, it should be '5,150' or '5 150'.

2. I found several typographical errors throughout the manuscript. For instance:
   - '5150m'. Please add a space between the number and the unit, like '5,150 m'.
   - Each number should have a standard format. Replace '570' with '570 mm'.
   - Replace 'Yellow River' with 'Yellow River Basin'.

3. Line 170: This paper provides the latest evidence of dramatic hydroclimatic changes in the headwaters of the Yellow River Basin (DOI:10.1088/1748-9326/ab9466).

4. Figure 1
   - Replace 'elevation' with 'DEM' as you use DEM in the figure caption. Otherwise, using elevation in the figure caption. Keep it consistent.
   - Part of the caption for subplot b is obscured.

5. Figure 3
   - In subplots d and e, there is a space between the subplot number and the caption. However, in subplots a, b, and c, there is no space between the number and the caption. Please carefully review the entire manuscript to avoid these occurrences.
   - Subplot e: place a space between the number and its unit.
   - Subplot f and g: consistently use a range of '0 to 700' in these two plots. The values in subplot f are almost reaching the top.
   - Subplot e: the ticks should align with the years instead of being placed in the middle of the years.
   - In the vertical axis titles of each subplot, there should be a space between the variable and the unit. This suggestion applies to all subsequent figures. For example, Replace 'temperature(℃)' with 'Air temperature (℃)'. Use 'Air temperature' because it occurred in the figure caption and elsewhere. Please note once you define the variable name, please consistently use it throughout the manuscript.

6. Figure 4
   - Figure 4 needs redesign. It is a little bit difficult to quickly understand the content of this figure according to the current information provided.
   - My understanding is that the first and second columns describe sub-diurnal and daily variations in four variables at the six sites, and the third and fourth columns describe similar variations at another six sites. If my understanding is correct, please clearly indicate that each subplot in the first and second columns shares the same legend, and

each subplot in the third and fourth columns shares another legend. Consider putting two legends at the top of the figure. This suggestion applies to all subsequent figures.

- I like the way to describe variations in Air temperature, subplot e-h consistently use the same range -20 – 20. Apply this to the other three variables. This suggestion applies to all subsequent figures.

7. Section '4 Potential applications enabled from this integrated dataset': Although the author discussed potential applications, more specific examples should be provided.

8. Overall, the manuscript requires careful reading and editing, including both textual content and figures. Language also requires polishing.

---

## Author Response (AR2)

**Response to the handling topic editor**

Dear Dr. Dalei,

We gratefully thanks for the precious time you have spent handling our manuscript. We have carefully considered the suggestions from the Reviewers and made reversions accordingly. Besides, the reference list style is updated following the style recommended by Copernicus. Our response to reviewers' comments is shown below.

We appreciate for Editor/Reviewers' warm work earnestly, and hope that these revisions successfully address their concerns and will meet with approval.

Once again, thank you very much for your comments and consideration of this manuscript.

Best regards,

Zhipeng Xie on behalf of all co-authors

**Response to referee report 1**

Thank you for the revisions made to the manuscript based on previous feedback. The manuscript has seen considerable improvement, but there are still areas that require further attention to ensure clarity and comprehensive presentation.

**Response:** We are grateful for your thoughtful feedback and suggestions, which have enabled us to make significant improvements to the manuscript.

1) L58-59: Consider adding specific impacts or key findings that underline the importance of this new dataset, not just use this simplified sentence.

   **Response:** The following revision has been made to the last two sentences in the abstract. Thank you very much for your suggestion.

   Line 58-62: With the greatest number of stations covered, the fullest collection of meteorological elements, and the longest duration of observation and recording to date, this dataset is the most extensive hourly land-atmosphere interactions observation dataset for the TP. It will serve as the benchmark for evaluating and refining land surface models, reanalysis products, and remote sensing retrievals, characterizing fine-scale land-atmosphere interaction processes of the TP, as well as underlying influence mechanisms.

2) L68-69: Specific the type of "evident changes" observed in the TP (e.g., temperature changes, precipitation patterns) to provide more context and clarity for the reader.

   **Response:** The following modification has been made to provide more context and clarity about the "evident changes" observed in the TP.

   Line 69-71: Many of the TP's environmental system components have experienced evident changes over the past few decades (Kang et al., 2010), for example, substantial retreat of glaciers (Yao et al., 2012), and dramatically decrease of snow depth and snow cover under the warming climate (Qin et al., 2006; You et al., 2020).

3) L113-114: It would be beneficial to add more discussions about existing datasets and their limitations to contextualize the necessity of the new dataset. This comparison will help in understanding the incremental or significant advancements made.

   **Response:** Yes, we totally agree with your suggestion on how to contextualize the necessity of the new dataset. That is the strategy we have employed to summarize and comment the existing publicly available field observation datasets in this part. We have made the following revision to clearly show the limitations of the existing datasets. Thank you very much.

   Line 114-117: A rising number of field observation datasets are progressively being made accessible and freely available to the public (e.g., Che et al., 2019; Ma et al., 2020; Liu, et al., 2023; Meng et al., 2023). However, to the best of our knowledge, thorough data quality control procedures have not been implemented to the current publicly available datasets for land-atmosphere interactions over the TP, although these datasets have contributed significantly to the study of climate and environmental change in the mountainous region.

4) The methodology section would benefit from more detailed descriptions of the statistical tests to analyze the data. Clarification on the choice of these methods and their appropriateness for

your dataset would strengthen this section.

**Response:** Based on your suggestion, we have made reversions on the original manuscript to clarify the choice of methods used in processing the dataset.

The following content is added to clarify the choice of the data quality control procedures used in this study. Line 257-258: The guidelines provide comprehensive documentation on basic quality control procedures used to ensure the accuracy of meteorological observations following World Meteorological Organization (WMO) standards.

To shed light on the rationale behind the usage of constant range thresholds, we include the following information when describing the constant limits used in range checks. Line 273-276: The range limits used in this study were constant, ignoring the seasonal variations, in contrast to the climatological limit check, where the range of limits depends on the season and regional climatic conditions (typically at least 20 years of archived data are required to define climate range thresholds).

The following modification has been made to provide more detail information on the temporal check. Line 284-287: It was discovered during testing that there would be a significant uncertainty if soil temperature, soil moisture, and surface radiations were subjected to temporal consistency test since the former two elements frequently exhibit little to no fluctuation but surface radiations are expected to exhibit substantial variability. As a result, we did not perform similar check for surface radiations, soil temperature, and soil moisture.

5) In Section 4 (L520-545), adding comparative analysis with prior data from other studies or contrasting regions will help in emphasizing the unique contributions of this new dataset.

**Response:** Following your suggestion, comparative analysis have been added to the original manuscript. Besides, a thorough reversion has been made to the potential applications of this dataset based on Reviewer 2, the updated content is listed as follows as well. Thank you very much.

Line 525-531: Compared with prior datasets released (e.g., Ma et al., 2020; Meng et al., 2023), this dataset involves more stations with various surface characteristics, allowing for a more thorough characterization of the spatial heterogeneity of land-atmosphere interactions over the TP. A strict data quality control procedure was implemented to process the dataset and quality flag was provided for each record, this pioneering initiative facilitates data users access to reliable observations, and minimizes the use of erroneous data, enables its widespread usage in studying the earth system of the TP.

Line 533-555: More specifically, field observations conducted across various landscapes and scales are indispensable for gaining a comprehensive understanding of the interactions between the land surface and overlying atmospheres. Taking the lake-atmosphere interactions as an example (e.g., Li et al., 2015; Wang et al., 2019), current field observations can provide fine-scale multi-component integrated observations at spatial and temporal scales ranging from centimeters to kilometers and from and from seconds to sub-hourly. By using the three-dimensional measurements from PBL tower, eddy covariance system, profile measurements of temperature, humidity, and wind by microwave radiometers, wind profiler, and radiosonde system, the physical mechanisms of land-atmosphere and boundary layer processes over the TP can be systematically investigated. Furthermore, field measurements are widely used to derive or calibrate land surface parameters for regional-scale estimate, satellite retrieval, and

numerical simulation of energy and water exchange over heterogeneous landscape (Yang et al., 2008; Chen et al., 2013). There is no doubt that this enhanced observation network enables a systematic assessment of model robustness and uncertainty in representing the land-atmosphere interactions in complex mountainous regions, providing better guidance for physical parameterization optimization of numerical models involving cryospheric, hydrologic, and atmospheric processes in the intricate TP terrain. Meanwhile, extensive field measurements are critical for validating, calibrating, and refining of remote sensing retrieval algorithms over the topographically complicated terrain. For instance, Yuan et al., (2021) used in-situ measurements from this dataset to present an enhanced canopy transpiration model as well as an improved approach for calculating soil evaporation using soil moisture and texture. Systemic biases in key land surface parameters in the reanalysis products can be decreased by integrating synthesis ground-based datasets and revised satellite products through sophisticated data assimilation techniques. For instance, Qi et al., (2023) improved the accuracy of land surface temperature retrieval over the TP based on the in-situ data. The combination of credible datasets provides multidimensional insights into the intricate mechanisms driving the recent changes across the fragile environments of the TP. This makes possible to comprehend the TP's critical role in Asian monsoons, water resources, and global climate teleconnections. In addition, predicting future changes and developing adaptive strategies for the environment and communities of the TP that are currently experiencing disproportionate climate change impacts depend on these integrated land surface observations.

6) The conclusion reiterates much of the information from the abstract and introduction and needs to be rewritten. To add value, consider highlighting the most advantages of this new dataset, the important findings from your dataset and specific implications for future research or policy changes prompted by this new dataset.
**Response:** We have made throughout reversion on the conclusion based on your suggestion. Thank you very much. The updated conclusion is listed as follows.
Line 569-581: This paper presents a suite of integrated field observations of land-atmosphere interactions over the TP with the cooperation of several agencies and organizations dedicated to field observations throughout the TP over several decades. This dataset includes hourly measurements of soil hydrothermal properties, near-surface micrometeorological conditions from 12 stations spanning up to 17 years (2005-2021). This paper highlights the complexity and spatial heterogeneity of land-atmosphere interactions over the mountainous region by describing in detail the observation network and presenting the hydrometeorological characteristics, soil hydrothermal properties, and surface energy balance components of these stations covering various landscapes over the decades. All of the data series in this dataset have been quality controlled using a combination of automatic error screening, manual inspection, diagnostic checking, adjustments, and quality flagging, as compared to other similar datasets that have previously been released. Suspicious and erroneous data were identified, and a QC code was assigned to each variable value. The specially designed data processing procedures tailored to handle the data issues of this integrated network is detailed described. It is indisputable that the long-term hourly quality-assured dataset presented here will contribute to a broad research effort and help advance the fine-scale understanding of the

land-atmosphere interactions over the heterogeneous TP region, refine land surface models, reanalysis products, and remote sensing retrievals.

**Response to referee report 2**

Most of the previous concerns have been adequately addressed. The following issues require attention:

1. Please ensure consistent use of scientific notation throughout the manuscript. For example, it should be '5,150' or '5 150'.
   **Response:** Modifications have been made based on your suggestion. Thanks.

2. I found several typographical errors throughout the manuscript. For instance:
   '5150m'. Please add a space between the number and the unit, like '5,150 m'.
   Each number should have a standard format. Replace '570' with '570 mm'.
   Replace 'Yellow River' with 'Yellow River Basin'.
   **Response:** We're grateful for your careful review of the manuscript. These errors have been fixed based on your suggestion.

3. Line 170: This paper provides the latest evidence of dramatic hydroclimatic changes in the headwaters of the Yellow River Basin (DOI:10.1088/1748-9326/ab9466).
   **Response:** Thanks for your recommendation, the paper has been added to the reference list.

4. Figure 1
   Replace 'elevation' with 'DEM' as you use DEM in the figure caption. Otherwise, using elevation in the figure caption. Keep it consistent.
   Part of the caption for subplot b is obscured.
   **Response:** Corrected as suggested.

5. Figure 3
   In subplots d and e, there is a space between the subplot number and the caption. However, in subplots a, b, and c, there is no space between the number and the caption. Please carefully review the entire manuscript to avoid these occurrences.
   Subplot e: place a space between the number and its unit.
   Subplot f and g: consistently use a range of '0 to 700' in these two plots. The values in subplot f are almost reaching the top.
   Subplot e: the ticks should align with the years instead of being placed in the middle of the years.
   In the vertical axis titles of each subplot, there should be a space between the variable and the unit. This suggestion applies to all subsequent figures. For example, Replace 'temperature(℃)' with 'Air temperature (℃)'. Use 'Air temperature' because it occurred in the figure caption and elsewhere. Please note once you define the variable name, please consistently use it throughout the manuscript.
   **Response:** Corrected as suggested except for the subplot f and g, where f shows the downward longwave radiation (with values greater than or equal to 0) and g shows the upward longwave radiation (values can be either positive or negative), so the range of '0-550' was used in f and '-100-650' was used in g. Thanks very much for your suggestion and understanding.

[Figure]

Figure 3

6. Figure 4

Figure 4 needs redesign. It is a little bit difficult to quickly understand the content of this figure according to the current information provided.

My understanding is that the first and second columns describe sub-diurnal and daily variations in four variables at the six sites, and the third and fourth columns describe similar variations at another six sites. If my understanding is correct, please clearly indicate that each subplot in the first and second columns shares the same legend, and each subplot in the third and fourth columns shares another legend. Consider putting two legends at the top of the figure. This suggestion applies to all subsequent figures.

I like the way to describe variations in Air temperature, subplot e-h consistently use the same range -20 − 20. Apply this to the other three variables. This suggestion applies to all subsequent figures.

**Response:** Thank you very much for pointing these issues out. Most of your concerns are addressed in the new figures (Figure 4, 5, 6, 7, 8). One thing that needs to be clarified here is that the consistency of the variation range between subplots has been considered as much as possible in our design of the measures of variation for each variable. However, because of the difference between the sub-diurnal and daily variation and the difference among stations (particularly the surface radiations and turbulent fluxes), it is challenging to distinguish the variations between stations when using the same range among the subplots as data points will overlap between sites. Our solution to this issue is to maximize the difference in the variations between stations by making some targeted adjustments to the variation range of variables with significant spatial difference (e.g., Rsu, and turbulent fluxes), while maintaining the

variation changes of other subplots as consistence as possible (e.g., Rsd, Rld and Rlu).

[Figure]

Figure 4

Figure 5

[Figure]

Figure 7

Figure 8

7. Section '4 Potential applications enabled from this integrated dataset': Although the author discussed potential applications, more specific examples should be provided.

**Response:** We have made a further reversion on the potential applications of this dataset, the revised content is listed as follows. We hope the modification made on the revised manuscript will cover the reviewer expectation.

Line 533-555: More specifically, field observations conducted across various landscapes and scales are indispensable for gaining a comprehensive understanding of the interactions between the land surface and overlying atmospheres. Taking the lake-atmosphere interactions as an example (e.g., Li et al., 2015; Wang et al., 2019), current field observations can provide fine-scale multi-component integrated observations at spatial and temporal scales ranging from centimeters to kilometers and from and from seconds to sub-hourly. By using the three-dimensional measurements from PBL tower, eddy covariance system, profile measurements of temperature, humidity, and wind by microwave radiometers, wind profiler, and radiosonde system, the physical mechanisms of land-atmosphere and boundary layer processes over the TP can be systematically investigated. Furthermore, field measurements are widely used to derive or calibrate land surface parameters for regional-scale estimate, satellite retrieval, and numerical simulation of energy and water exchange over heterogeneous landscape (Yang et al., 2008; Chen et al., 2013). There is no doubt that this enhanced observation network enables a systematic assessment of model robustness and uncertainty in representing the land-atmosphere interactions in complex mountainous regions, providing better guidance for physical parameterization optimization of numerical models involving cryospheric, hydrologic,

and atmospheric processes in the intricate TP terrain. Meanwhile, extensive field measurements are critical for validating, calibrating, and refining of remote sensing retrieval algorithms over the topographically complicated terrain. For instance, Yuan et al., (2021) used in-situ measurements from this dataset to present an enhanced canopy transpiration model as well as an improved approach for calculating soil evaporation using soil moisture and texture. Systemic biases in key land surface parameters in the reanalysis products can be decreased by integrating synthesis ground-based datasets and revised satellite products through sophisticated data assimilation techniques. For instance, Qi et al., (2023) improved the accuracy of land surface temperature retrieval over the TP based on the in-situ data. The combination of credible datasets provides multidimensional insights into the intricate mechanisms driving the recent changes across the fragile environments of the TP. This makes possible to comprehend the TP's critical role in Asian monsoons, water resources, and global climate teleconnections. In addition, predicting future changes and developing adaptive strategies for the environment and communities of the TP that are currently experiencing disproportionate climate change impacts depend on these integrated land surface observations.

8. Overall, the manuscript requires careful reading and editing, including both textual content and figures. Language also requires polishing.

   **Response:** The revised manuscript was carefully checked and edited for the English language to take into account all suggestions by the reviewer.

**Response to referee report 3**

In the early stages of the manuscript, I had three main concerns, all of which were addressed by the authors in this round of revisions.

I originally felt that the introduction was too long (Lines 64-155). It has since been updated (Lines 63-142) and it still follows a logical framework needed to introduce the goal/purpose of the study. I also felt that Section 2 (Observation Network and Data Processing) lacked the necessary details needed for a scientific data paper. The authors made significant changes to Section 2.3 (Data Post-Processing Workflow) which helps the reader to better understand the processing workflow and what QA/QC processes actually took place. Lastly, I had questions/concerns about the validity of the EC data and was curious why the heat fluxes were identified to be so 'bad' (Figure 8 and Figure B3) at each site. The authors mentioned that they read/plotted the incorrect data in that figure, and while it has been fixed, the updated figure (to a much lesser degree) still has some 'bad' latent heat flux and sensible heat flux data (which is a bit surprising). Figure 8 also appears to be corrected. Overall, I am satisfied with the revisions that the authors made for my comments and for the other reviewer comments. Following a detailed proofread (and final review by editor), I believe the manuscript can be accepted.

**Response:** We would like to thank you for your careful reading, helpful comments, and constructive suggestions, which has significantly improved the presentation of our manuscript.

---

## Author Response (AR3)

**Response to the topic editor**

Dear Dr. Dalei,

We appreciate you posting your concern about the figures. We have checked the figures and revised the Figure 1 and 9 based on your suggestion.

In the updated Figure 1, we revised the colors scheme with the adoption of a colorblind-friendly palette to better illustrate the spatial distribution of plant functional types (PFTs). In addition, the font size is also modified.

In the updated Figure 9, we used a different color scheme and revised the legends.

To meet the requirements for publishing in ESSD, we thoroughly examined the color scheme for the remaining figures using the Coblis – Color Blindness Simulator, which was recommended by the editorial board.

Thank you very much for your comments and consideration of this manuscript.

Best regards,

Zhipeng Xie on behalf of all co-authors

[Figure]

Updated Figure 1

[Figure]

Updated Figure 9